

# Real time analysis of insoluble particles in glacial ice using single particle mass spectrometry

Matthew Osman[1], Maria A. Zawadowicz[2], Sarah B. Das[3], Daniel J. Cziczo[2,4]

[1] MIT/WHOI Joint Program in Oceanography/Applied Ocean Sciences and Engineering, Woods Hole Oceanographic Institution, Woods
Hole, MA, 02543, USA

[2] Department of Earth, Atmospheric, and Planetary Sciences, Massachusetts Institute of Technology, Cambridge, MA, 02139, USA

[3] Department of Geology and Geophysics, Woods Hole Oceanographic Institution, Woods Hole, MA, 02543, USA

[4] Department of Civil and Environmental Engineering, Massachusetts Institute of Technology, Cambridge, MA, 02139, USA

*Correspondence to*: Daniel J. Cziczo (djcziczo@mit.edu)

**Abstract.** Insoluble aerosol particles trapped in glacial ice provide insight into past climates, but analysis requires information on climatically-relevant particle properties, such as size, abundance, and internal mixing. We present a new analytical method using a time-of-flight single particle mass spectrometer (SPMS) to determine the composition and size of insoluble particles in glacial ice over an aerodynamic size range of ~0.2 – 3.0 µm diameter. Using samples from two Greenland ice cores, we

developed a procedure to nebulize insoluble particles suspended in melted ice, evaporate condensed liquid from those particles, and transport them to the SPMS for analysis. We further determined size-dependent extraction and instrument transmission efficiencies to investigate particle-class specific mass concentrations. We find SPMS can be used to provide constraints on the aerodynamic size, composition, and relative abundance of most insoluble particulate classes in ice core samples. We

describe the importance of post-aqueous processing to particles, a process which occurs due to nebulization of aerosols from an aqueous suspension of originally soluble and insoluble aerosol components. This study represents an initial attempt to use SPMS as an emerging technique for the study of insoluble particulates in ice cores.

## 1. Introduction

Aerosol particles play a significant role in global climate, both directly, through regulation of regional and global radiation budgets and atmospheric photochemical cycles, and indirectly, through their ability to modulate cloud microphysical and precipitation processes. In both cases, the abundance,



size, morphology, and composition of the aerosols dictate their ability to affect climate (Seinfeld and Pandis 2006). As such, the ideal atmospheric particle measurement has been described as one that is able to "count each particle and report its size, chemical composition, and morphology rapidly" (Wexler and Johnston, 2011).

Inferences of past climates have been achieved using ice core records, whose chronologies provide direct and high-resolution records of past atmospheric composition (Legrand and Mayewski, 1997). A large number of methodologies have been developed to determine properties of aerosols in snow and ice. However, all have important limitations.

So-called "offline" particle retrieval methods, requiring a precursor step to first concentrate
particles prior to analysis, have been widely employed for ice core studies. Offline studies typically use a pre-existing analytical instrument and are therefore of low expense, requiring only that the sample be transported to the instrument. When paired with quantitative analytical techniques, such as Raman (Sakurai et al., 2011) and sublimation energy dispersive (SED) X-ray spectroscopy (Drab et al. 2002, Iizuka et al., 2009, Oyabu et al. 2015), as well as optical techniques, including transmission or scanning
electron microscopy (Murr et al. 2004; Ellis et al. 2015), offline methods can achieve determinations of particle size, composition, and morphology. There are limitations of offline techniques, however, including the time and resources required for sample recovery and transport, and these limitations often render such methods insufficient for acquiring contiguous, high-resolution ice core particle records. Further, offline methods can increase the chance of sample contamination and/or particle losses
occurring during handling, transport, and storage (Ohata et al., 2011).

"Online" instrumentation, in contrast, allows for rapid, real time delineations of ice core particulate properties, necessary for contiguous, high-resolution measurements of particle mass-concentrations and (or) size-distributions along an ice core. For example, Coulter Counter (CC) or laser particle detection (LPD) based instrumentation is typically used for determining insoluble particle
("dust") size distributions in glacial ice (Hamilton and Langway, 1967; Knipperz and Stuut, 2014). However, neither CC nor LPD techniques provide associated information on particle composition or morphology; broad assumptions of both parameters are thus required to infer ice core dust concentrations from these instruments. Alternate online techniques, such as inductively coupled plasma





mass spectrometry (ICP-MS) and ion chromatography (IC), are also routinely used to determine elemental (e.g., Cd, Ce, Zn, Pb) and soluble ion (e.g., $Na^+$, $SO_4^{2-}$, $NH_4^+$) concentrations in ice cores, providing independent indices of past dustiness (e.g., Rhodes et al. 2011), and anthropogenic particle emissions (e.g., Osterberg et al. 2008, McConnell et al. 2008). Albeit analytically precise, ICP-MS and

IC rely on techniques that analyze only a portion of aerosols in an aqueous suspension/solution, and not the properties of the progenitor aerosols themselves, i.e., particulate morphology or size.

The recent advent of a commercially available intracavity laser-induced incandescence photometer, the Single Particle Soot Photometer (SP2; Droplet Measurement Technologies, Inc. Boulder, CO) has allowed for a new class of online measurements where both concentrations and size

distributions of (refractory) carbonaceous particles in snow and ice can be recovered on a per-particle basis (McConnell et al. 2007; Kaspari et al. 2011; Ginot et al. 2014; Sun et al. 2014). McConnell et al. (2007) were the first to measure past black carbon deposition continuously along a Greenlandic ice core using SP2 instrumentation. More recent SP2 studies have highlighted the necessity of quantifying instrumental errors and systematic uncertainties when extracting trace amounts of highly heterogeneous

material from snow and ice (Schwartz et al. 2012; Ohata et al. 2013; Wendl et al. 2014). Specifically, these studies have shown that correction factors, primarily a result of the efficacy achieved in nebulizing insoluble particles from a melted ice core matrix and aerodynamically delivering them to the SP2 for analysis, can be large and vary significantly between laboratory set-ups (Lack et al., 2014).

To date no single online method has proven capable of determining the size and chemical

composition for all particle-types situated in glacial ice. Here we expand upon the advantages of the SP2-based methodology by using a time-of-flight (ToF) single particle mass spectrometer (SPMS), allowing for analyses of small aerosol masses (~fg) and differentiation of internally mixed components – from both refractory (e.g. black carbon, mineral dust) to volatile (e.g. sulfuric acid, organic compounds) – on a particle-by-particle basis. This technique leverages a state of the art instrument used

for atmospheric studies (Murphy, 2005; Cziczo et al. 2006) to provide real-time (i.e., online) measurements, thereby minimizing contamination, losses during handling, and sample preparation requirements. We describe the development and application of the SPMS method to quantitatively resolve the size, abundance, and composition of a host of insoluble particle-types including, but not





limited to, mineral/metallic, biomass burning, soot and organic particles contained within ice cores. To demonstrate the utility of SPMS as a new online tool for qualitative and quantitative particulate measurements in glacial ice, we present data from discrete glacial samples from two sites in west-central Greenland. We conclude with suggestions for future application and improvements to the
method.

## 2. Methodology

### 2.1 Particle Analysis by Laser Mass Spectrometry (PALMS) and particle classification

The PALMS instrument has previously been used for *in situ* measurements of airborne particles in both laboratory settings, as well as airborne and ground-based field campaigns. The PALMS
instrument has been described previously (Murphy and Thomson, 1995; Thomson et al., 2000; Cziczo et al., 2006, Cziczo et al., 2013). In brief, accumulation- to coarse-mode (~0.2 − 3.0 µm aerodynamic diameter) particles entering the PALMS inlet are brought into a vacuum and passed through two continuous Nd-YAG lasers causing a scattering of green-wavelength light ($\lambda \approx 532$ nm). The time difference between, and the magnitude of, scatter signals provide a measure of the particle diameter
(aerodynamic and geometric, respectively). The scattering additionally triggers a 193nm excimer laser pulse (~10ns pulse duration; $10^9$ W cm$^{-2}$), which desorbs and ionizes the particles. The resultant ions are then passed to a ToF mass spectrometer, providing a mass spectrum of individual particles in real time. PALMS produces mass spectra for >95% of particles detected by the visible laser (Cziczo et al., 2006).

Since PALMS has a single ToF mass spectrometer, spectra are limited to either positive or
negative ions during a given sampling period (Murphy and Thompson et al. 1997a,b). The classification of a particle in either the positive or negative ion mode depends on that particle's chemical makeup, e.g., a mineral dust particle identified in the positive ion mode is typically associated with metal-oxide ($Na^+$, $K^+$, $Al^+$, $Fe^+$, $FeO^+$) markers. An empirical algorithm (employed in the *Interactive Data Language, IDL*), tuned to previously acquired laboratory and field data for ambient atmospheric
particles (Cziczo et al., 2013), was used to automate particle classification based on key markers of each particle's mass spectrum. Particle classifications derived using this algorithm are described in Tables 1 and 2, and include mineral dust, biomass burning particles, soot (elemental carbon), organic and/or





sulfate/organic material, heavy oil/combustion products, and sea salt. Due to deviations of particles found in ice core samples from those in an airborne state (Murphy and Thomson, 1997a,b; Cziczo et al., 2013) additional algorithms were developed to distinguish the following classes: (1) calcium-rich  (2) "processed" (i.e., containing a broad mixture of both refractory and volatile components), (3)

phosphorous-rich biological and (4) phosphorous-rich inorganic particles (Zawadowicz et al., 2016). Additional information on the processing of atmospheric particles trapped in ice is provided in Sect. 2.3.1, and the methodology used for elimination of contamination particles (a result of ice processing) is given in Sect. 2.2.2.

## 2.2 Samples

**2.2.1 Ice core samples**

The ice core samples used in this study are from two distinct locations in west-central Greenland. DS14 is situated on the Disko Island ice cap (69˚39'N, 52˚44'W) at an elevation of ~1250 m asl. ~20 km from the coast, and represents a "coastal" site. GW14 is located on the Greenland ice sheet (69˚12'N, 44˚31'W) at an elevation of ~2250 m, ~240 km from the coast and represents an "inland" site.

Average accumulation rates, calculated from age-depth and density profiles, are high at both locations: ~0.5-0.6 m water equivalent (w.eq.) yr$^{-1}$ and ~0.3-0.4 m w.eq. yr$^{-1}$ at DS14 and GW14, respectively, allowing for seasonally-resolved age-depth profiles (accuracy for GW14 is estimated to be within 2-3 months; accuracy at the higher melt intensity DS14 site is estimated to be within ± 1 years) using seasonal maxima in water-isotopic and soluble ion chemistry measurements. Samples from both sites

are from the modern (last few years) era.

Core processing involved discretization of samples via stainless steel band saw cuts followed by standard sample preparation procedures for firn samples (Osterberg et al., 2006), which included manually shaving the outer 4-5 mm of firn/ice from each sample using a pre-cleaned ceramic ZrO microtome blade under a laminar flow clean bench. All samples remained chilled at -20˚C (diurnal

variation <± 5˚C) from collection until ~2 hours prior to PALMS analysis, minimizing potential volatile losses (Ohata et al., 2013, Wendl et al., 2014). Four samples were analyzed from GW14 (GW14-01 to -04), sampled discretely at 20 cm intervals between depths of 4-5 m. Each sample corresponded to ~2-3



months over winter 2004 – fall 2005. Six samples were analyzed from DS14 (DS14-01 to -06) sampled discretely at 12.5 cm intervals between depths of ~2-3m, corresponding to the year ~2011 ($\pm$1 year). For the purposes of this study, all samples from each site are presented as compiled results in Sect. 3.2, except DS14-05 (see discussion in Sect. 3.2.3) and DS14-06 (see discussion in Sect. 3.2.2).

### 2.2.2 Blank tests and insoluble artifacts

To delineate ZrO ceramic or metal (stainless steel) artifacts derived from core processing, band saw cuts were made on frozen ultrapure water controls ($18.2\ M\Omega$) and processed as described in Sect. 2.2.1 to infer baseline levels of metal and ZrO contamination. Stainless steel has been observed with PALMS previously and the signature of this material is therefore well known (Murphy et al., 2010). Stainless steel contamination was identified as $Fe^+$, $Al^+$ and $Mo^+$ with occasional $Sn^+$ and $Ti^+$ and/or $TiO^+$ without the presence of alternate mineral/metallic markers (Table 1). Blank tests indicated negligible contamination of stainless steel from core processing when band saws were used ($\ll1\ sec^{-1}$). Contamination artifacts of ZrO-rich particles from the microtome ceramic blade were more common. Due to the low probability of natural stainless steel and ZrO-enrichment (Thomson et al., 1997a,b), these particles were assumed to be indicative of contamination and eliminated from the data.

### 2.3 Nebulization

Nebulization was used to aerosolize particles after melting the ice samples. The condensed-phase water was then evaporated in the low-humidity nebulization flow, thereby releasing particles. In order to quantitatively analyze the production of aerosol, the liquid solution was nebulized and transported to PALMS at known flow rates. Figure 1a shows a schematic of the experimental setup. Melted sample was placed into a glass container attached to a custom Collison-type atomizer. Dry, inert carrier gas ($N_2$) was separated into two flows using a rotameter, creating "wet" and "dry" flows, 2.0 L $min^{-1}$ and 3.0 L $min^{-1}$ ($\pm$ ~0.1 L $min^{-1}$), respectively. The former flow was directed to the atomizer, dispersing the liquid sample into a mist with insoluble particles contained within a fraction of the droplets. The atomized particles were adjoined with the dry flow, which dropped the relative humidity and caused evaporation of residual condensed water prior to entering the PALMS inlet (i.e., PALMS





analyzed dry residual particles). The PALMS inlet accommodates a flow of 0.44 L min$^{-1}$ so the balance of the total flow is filtered and returned to the atmosphere. Samples GW14-01 to GW14-04 and DS14-01 to DS14-04 were analyzed with PALMS in positive and negative ion mode alternately for 10 minutes resulting in a frequency of ~1-2 particles sec$^{-1}$. Sample DS14-05 was analyzed for one hour

while DS14-06 was used for additional solubility tests (Sect. 2.3.1). Between runs, the nebulizer and sampling beaker were cleaned and sonicated for 15 minutes using ultrapure (Milli-Q; 18.2 MΩ) water.

**2.3.1 Soluble artifacts**

An important aspect of analyzing particles re-aerosolized from a liquid suspension, as described in the last section, is the homogenous dilution of soluble components of the original aerosol in the liquid

water matrix. Following evaporation of the condensed water during nebulization and transport (Fig 1a), this material is formed from two distinct processes during nebulization and transport to PALMS. First, nebulized droplets not containing an insoluble residual particle evaporate to form small particles comprised solely of the solute ions in the original droplet. Second, droplets containing insoluble residual particles are "coated" with the soluble ions following evaporation of the droplet's condensed

liquid. This may be visualized as a slurry of insoluble mineral dust particles in a water matrix containing a dissolved salt. Nebulized droplets of the slurry may or may not contain a mineral dust particle with the frequency dependent on the concentration of the dust; however, all droplets will contain salt ions with the amount dependent on the concentration in the solution. In this simple system, nebulization would produce either pure salt particles or mineral dust particles coated by salt. We refer to

the latter as *post-aqueous coating*. This process is illustrated in Figure 1b.

Residual particles composed purely of soluble material after removal of condensed phase water were not typically observed. This is likely because the amount of soluble material in ~μm diameter droplets produced by the nebulizer is not high enough to produce particles above the lower PALMS detection limit of ~0.2 μm (i.e., the concentration is <1% by volume with respect to water). To quantify

the degree of post-aqueous processing, three experiments were performed: 1) sample DS14-06 was analyzed with PALMS in the negative ion mode to determine background particulate levels, 2) the sample was then filtered through 0.02μm inorganic membrane filters (Whatman Anotop 10) to remove





all particulates in the PALMS size range and 3) the filtered sample was then doped with 746 nm diameter polystyrene-latex spheres ("PSLs"; Polyscience, Inc, Warrington, PA, USA) to provide a single core of a specific chemical composition on which soluble material would coat. For experiment 2) PALMS exhibited a transmission of <<1 sec$^{-1}$ (i.e., less than 1% the rate without removing insoluble

particles from the ice), evidence that droplet residuals were not of sufficient size for detection. Quantification of the post-aqueous coating is discussed further Sect. 3.2.2.

### 2.3.2 Calibrating particle extraction efficiency

A particle extraction efficiency curve was developed to quantitatively determine the number of particles analyzed by PALMS versus the initial number concentration of particles in solution. The

extraction efficiency, $\varepsilon$, represents the multiplicative probability that a particle of diameter $D_p$ will be (1) nebulized, (2) transported to the PALMS inlet, (3) detected and (4) ablated and ionized with the excimer laser. Both $\varepsilon$ and PALMS transmission have previously been shown to be a function of aerodynamic particle diameter (Cziczo et al., 2006) so a size-dependent extraction efficiency curve in the particle diameter, $D_p$ = ~0.2 - 3.0 μm was determined here with monodisperse PSL particles of

known number concentrations via the following:

$$\varepsilon_{PSL}(D_p) = \frac{n_{PALMS}(D_p) \cdot F_{flow}}{m_{PSL}(D_p) \cdot V_{neb}},$$ (1)

where,

$$n_{PALMS}(D_p) = \frac{f_{PALMS}(D_p)}{F_{inlet}}$$

where $F_{flow}$ is the gas flow rate (monitored continuously using a *MesaLabs Bios DryCal 220*; STP cc sec$^{-1}$); $V_{neb}$ is the rate of liquid nebulization (determined with a scale; $4.4 \cdot 10^{-6} \pm 1.6 \cdot 10^{-6}$ mL sec$^{-1}$);

$m_{PSL}(D_p)$ is the liquid number concentration (particles mL$^{-1}$); $n_{PALMS}(D_p)$ is the number concentration of particles analyzed by PALMS per unit volume air (particles cc$^{-1}$), defined as the frequency of PSL analyzed by PALMS, $f_{PALMS}(D_p)$ (particles sec$^{-1}$), normalized by the PALMS inlet flow rate, $F_{inlet}$, (STP cc sec$^{-1}$). The size dependent efficiency was assumed to be independent of particle composition and aerodynamic shape, i.e., $\varepsilon_{PSL}(D_p) = \varepsilon_{Particle}(D_p) = \varepsilon(D_p)$.



### 2.3.3 Polystyrene Latex Sphere (PSL) standards

The determination of $\varepsilon(D_p)$ from eq. (1) required particles of known size. Nine PSL sizes were used: 244, 341, 505, 652, 746, 953, 1500, 1990, and 3000 nm diameter, all NIST certified at 1% solid weight percent by volume (w/v) (Polyscience, Inc., Warrington, PA, USA). All PSLs were spherical

and near unit density (1.05 g cm³). The manufacture's weight percentage claim (1% w/v) was assumed sufficient for calculations (Wendl et al., 2014). Using this mass loading fraction, the 746 nm PSL was diluted with ultrapure water (Milli-Q; 18.2 MΩ) until a PALMS transmission response of 3-5 sec⁻¹ was reached. This corresponded to a number concentration of ~8.8·10⁶ particles cm⁻³. This standard number concentration was then used as the reference dilution ratio for the remaining PSL standards.

### 2.3.4 Mass concentration determination

A mass concentration can be determined with a statistically representative number of particles analyzed by PALMS. Here we define a statistically representative number of particles as the minimum number required to develop a particle size distribution such that outliers, particularly large particles, do not apply erroneous weighting to subsequent mass determinations. In the context of incandescence-

based single particle methods (i.e., SP2), Schwarz et al. (2012) have suggested 10,000 particles per sample.

As defined by Wendl et al. (2014), an "external" calibration approach, employed here, assumes that $\varepsilon$ for a given monodisperse PSL standard scales with an unknown, polydisperse ice core sample. By rearrangement of eq. (1):

$$m_{sample}(D_p) = \frac{n_{PALMS}(D_p) \cdot F_{flow}}{\varepsilon(D_P) \cdot V_{neb}} \qquad (2)$$

Incorporating continuous measurements of the composite flow balance, eq. (2) indicates that determination of the number concentration, $m_{sample}(D_p)$, of a sample containing particulates is feasible using PALMS. Note the external calibration approach does not account for drift in nebulization efficiency via eq. (1), which accumulates uncertainty over time. This suggests recalibration should be

executed periodically based on performance drift.

The normalized logarithmic size distribution of retrieved particles, i.e., those measured by





PALMS, for the $i^{th}$ particle class, can be described by:

$$N_{sample}^i(D_p) = \int_{LB(D_p)}^{UB(D_p)} \frac{F_{flow}}{\varepsilon(D_P)\cdot V_{neb}} \cdot \frac{d\,n_{PALMS}(\log D_p)}{d\,\log D_p} d\log D_p \qquad (3)$$

where the efficiency corrected size distribution of particles is integrated under the size-dependent upper to lower transmission/nebulization efficiency bounds UB($D_p$) and LB($D_p$), as determined in 2.3.2.

Assuming a spherical shape of all particles, the integrated mass, $M_{sample}^i(D_p)$, of a lognormal size distribution can be calculated by scaling the lognormal volume distribution (Seinfeld and Pandis, 2006) by a particle density representative of that particle class ($\rho^i$):

$$M_{sample}^i(D_p) = \rho^i \cdot \frac{\pi}{6} D_p{}^3 \int_{LB(D_p)}^{UB(D_p)} \frac{F_{flow}}{\varepsilon(D_P)\cdot V_{neb}} \cdot \frac{d\,n_{PALMS}(\log D_p)}{d\,\log D_p} d\log D_p \; = \rho^i \cdot \frac{\pi}{6} \cdot D_p{}^3 N_{sample}^i(D_p)$$

$$(4)$$

Due to the heterogeneous chemical makeup of individual particles, even within a single particle-class, a class-representative $\rho^i$ is subject to some uncertainty. Past studies have illustrated that PALMS can obtain error inclusive, objective estimates of $\rho^i$ (Murphy et al., 2004), provided a large number of particle spectra per class (nominally, >10,000).

## 3 Results and discussion

### 3.1 Size-dependent extraction and transmission efficiency

The size dependent extraction efficiency curve determined from the experimental data is illustrated in Figure 2. The 657 nm diameter particle size showed the greatest efficiency, determined via eq. (1) to be $4.0\cdot10^{-3}$ (0.40%). At the min and max PSL sizes, 244 and 3000 nm, respectively, the extraction efficiency decreases to $8.6\cdot10^{-5}$ and $8.3\cdot10^{-6}$, respectively (~1 particle out of ~$10^4$ or $10^5$). A

nonlinear least squares approach was used to fit a lognormal distribution to the nine PSL sizes, giving:

$$\varepsilon(D_P) = \frac{-0.017}{2\pi^{\frac{1}{2}}\log(0.57)} \exp\left(-\frac{[\log(D_P)-\log(0.66)]^2}{2\log(0.57)^2}\right) \qquad (5)$$

with a goodness-of-fit statistic (mean-square error) of $1.80\cdot10^{-7}$. The computed upper and lower 95% confidence intervals, calculated using an assumption of normally distributed model-observation residuals, encapsulate all data points (Figure 2).



The combined flow rate (i.e., the wet plus dry lines, $F_{flow}$ of eq. 1) was set to 5.0 L min$^{-1}$: a dry to wet-flow ratio of ~3:2 was experimentally found to evaporate the condensed aqueous sample from atomized particulates prior to reaching the PALMS aerodynamic inlet. Once nebulized, most particle losses are assumed to occur via "dumping" of excess flow (Figure 1a). Since the PALMS aerodynamic

inlet allows a maximum inlet flow rate of 0.44 liters per minute, an inferred ~91% of nebulized particles are lost to this flow. Scaling the efficiency curve by the ratio of excess-flow to the PALMS inlet flow provides a theoretical maximum nebulization efficiency of 0.045 (4.5% efficiency) at the 657 nm particle size, indicating that a non-negligible portion of the remaining atomized particles are also lost during transport to the PALMS inlet.

To determine the PALMS transmission efficiency, $\varepsilon_{trans}$, a monodisperse particle-laden PSL flow was split between PALMS and an optical particle sizer (OPS; *MesaLabs Bios DryCal 220*) and calculated as the flow-weighted ratio of the airborne number concentration of particles successfully ionized by PALMS (Figure 1) to that simultaneously measured by the OPS. The experimentally derived $\varepsilon_{trans}$ showed a maximum of ~15-16% efficiency over the particle size range ~300-1000 nm (Figure 2),

dropping to ~2% at ~250 nm and ~1% transmission at 3000 nm. The maximum $\varepsilon_{trans}$ determined here was ~5-6% higher than found by Cziczo et al. (2006). The differences are attributed to adjustments made to the PALMS aerodynamic inlet and sizing lasers since that study.

The methodology presented here was constructed to allow for a broader size distribution of aerosols sent to PALMS compared to SP2 nebulization techniques used in prior ice core studies

(McConnell et al., 2007; Ohata et al., 2011, 2013; Schwarz et al., 2012; Wendl et al., 2014) but with a lower extraction efficiency. For example, Ohata et al. (2011) achieved a max nebulization efficiency of ~12%, comparable to Schwarz et al. (2012) and Wendl et al. (2014). However, the SP2 achieves optimal transmission in the low-mid accumulation mode particle size (<400-500 nm; Schwarz et al., 2012), which have lower impaction probabilities (i.e., wall losses during transmission). As such, these

studies have commonly incorporated a U5000AT+ ultrasonic nebulizer (Cetac Technologies, Inc., Omaha, NE, USA), which reaches highest nebulization efficiency in the band 100 > $D_p$ > 500 nm (Schwarz et al., 2012); such a range would prove too restrictive for SPMS applications, which can detect >3000 nm diameter particles.





It is noted that achieving too high of an extraction efficiency could be disadvantageous for SPMS, should the number of particles reaching the SPMS inlet exceed that instrument's max transmission rate (~10 particles sec$^{-1}$ for PALMS; Cziczo et al., 2006) where the limit is data writing and laser repetition rate. This does not affect SP2, which can more rapidly measure the incandescence

5  of carbonaceous material passing through a continuous laser (i.e., 2-3 orders of magnitude higher), though with cost of not delivering information on particulate mixing state. More efficient nebulization at relevant SPMS sizes, coupled to more rapid excimer lasers and data writing, would increase the data acquisition rate.

### 3.2 Method application: Greenlandic ice core particulates

10  ### 3.2.1 Ice core particulate chemical classifications

4931 spectra (2362 positive and 2569 negative) were analyzed from DS14 (DS14-01 to DS14-05), and 553 spectra (233 positive and 320 negative) from GW14 (GW14-01 to GW14-04). The main discrepancy between the number of particles measured between the two sites is not particle loading, but due to 60 minutes of sampling time for DS14-05 (as opposed to 10 minutes for all other samples) in the

15  positive and negative ion modes. Categorization is broadly divided into natural and anthropogenic sources, as previously noted in modern-era particles from Greenland (Drab et al., 2002, VanCuren et al., 2012), but with some overlap. For example, biomass burning and mineral dust now come from both source-types. Representative particle spectra for all classes, in both the positive and negative ion modes, are provided in the Appendix.

20  In positive ion mode, mineral dust is distinguished using primary alkaline markers, Na$^+$, K$^+$, as well as Al$^+$, Fe$^+$ and other metals and their oxides. A Ca-rich particle sub-class of the mineral/metallic class was delineated by the additional presence of calcium and its oxides (e.g., Ca$^+$, CaO$^+$, CaOH$^+$ and Ca$_2$O$^+$ and/or CaKO$^+$). This mineral sub-class may represent a terrestrial carbonate dust source (Murphy and Thomson, 1997b) or marine-derived CaCO$_3$ or calcium-sulfates. We suggest the latter possibility is

25  consistent with the high prevalence of Ca-rich particles at the coastal DS14 site (~30% of positive particles) relative to the inland GW14 site (~8%; Figure 3). A fraction of the mineral/metallic-rich particles were mixed with organic components (e.g., carbon cluster isomers, C$_n^+$, hydrocarbons, organo-



nitrogen markers). This particle sub-class was delineated as "processed", as it is likely affected by post-aqueous processing; at this time we cannot fully decouple this artifact from an original atmospheric association (see also Sect. 3.2.2. below).

The biomass-burning category contains signatures of carbon-cluster isomers, elevated $K^+$ and/or

potassium sulfates ($K_3SO_4^+$). These particles were the second most abundant positive-ion mode type at DS14 (31%) with lower representation at GW14. In the negative ion mode, sulfate-organic rich spectra made up the largest proportion of particles at DS14 (>66%), but comprised just over one third of particles at GW14 (Figure 3). Inspection of the organic/sulfate class at both sites indicated that a substantial proportion of particles also contained elevated $PO_2^-$ and $PO_3^-$. Past studies have noted the

challenge in identifying the origin of P-rich particles, with both mineral and biological origin possible (Creamean et al., 2014). The classification scheme by Zawadowicz et al. (2016) was adopted here to identify the P-rich particle subset as either of biological or inorganic origin (Table 2), with the latter suggestive of mineral dust or fly ash particles. The biological particles were found to be largest contributor of particles analyzed at DS14 in negative ion mode, 39%. These particles were also found to

have the largest size of all the particle classes at both sites, with a median geometric diameter of 705 nm. This is consistent with a local marine biogenic source (Figure 4). Additionally, we note that biomass burning and biogenically-sourced particles can appear similar in the positive ion mode, especially if post-aqueous processing is involved (Zawadowicz et al., 2016). Due to similarities in particle size (median particle diameter within ~10 nm; Figure 4) and relative abundance with the

biological class, we interpret the positive ion mode biomass-burning particles to be of equivalent marine-biological origin at the DS14 site.

Soot is distinguished in both the positive and negative ion modes by the presence of elemental carbon (e.g., $C^+$, $C_2^+$, $C_3^+$, etc., or $C^-$, $C_2^-$, $C_3^-$, etc.) with organic peaks. Soot was common at both sites in the negative ion-mode, showing an abundance (composite between positive and negative spectrum)

of ~18% and 28% at DS14 and GW14, respectively. Both abundances are considerably higher than previously reported by Drab et al. (2002) in a firn sequence at Summit, Greenland, who estimated <5% of particles were soot or combustion-based although the authors noted the difficulty in optically evaluating these particles, due their tendency to aggregate. A currently unknown feature of the data is



the high negative polarity soot abundance at the GW14 site that appears to correspond to positive polarity mineral or metallic particles. This feature may be indicative of an anthropogenic particle influence. The heavy oil combustion class, distinguished by elevated vanadium markers ($V^+$ and $VO^+$), was of low abundance at both sites, representing <0.1% of all particles measured (not shown in Figure 3).

### 3.2.2 Post-aqueous processing

The inclusion of sea salt in both the positive and negative ion modes suggests an artifact arising from post-aqueous processing. This class was observed at both GW14 and DS14 but at low abundance (<1-2%). Sea salt aerosol would have dissolved upon melting of the ice samples prior to analyses. This classification may therefore represent evaporation of an uncommonly large droplet or a coating where the underlying particle composition is not observed. In addition, a non-negligible percentage of the positive-ion mode particles from both sites were not readily characterized, with "unclassified" particles comprising 8% and 6% of particles at DS14 and GW14, respectively. Manual inspection of spectra from this class indicates particles containing key markers from numerous classes (Cziczo et al., 2013), thereby leading to confusion of the classification method. This may represent particles undergoing severe post-aqueous processing or coagulation within the solution or after nebulization and led to the work discussed in more detail in the next section.

We note that a single-particle mass spectrometer similar to PALMS was recently used to measure chemical composition of insoluble particles in precipitation samples collected in Sierra Nevada (Creamean, et al. 2013, 2014, 2016). In those studies, post-aqueous processing was not investigated in detail and single particle classifications were not adjusted based on the possibility of processing.

### 3.2.3 Solubility tests

Atmospheric particles are not chemically homogenous and often contain both soluble and insoluble components (Murphy, 2005). Several particle classes presented here, specifically the sea salt and organic/sulfate classes and, to a lesser extent, the P-rich and Ca-rich classes, were inferred to contain an appreciable proportion of soluble components, though of indeterminate origin (i.e., of atmospheric or post-aqueous processing origin). To test the degree of post-aqueous coating to these four



particle classes, four negative ionic species were chosen as class-representative markers: the ions $PO_2^-$ + $PO_3^-$ (m/z = 63 + 79) for P-rich biological particles, $SO_3^-$ + $HSO_4^-$ (m/z = 80 + 97) for sulfate-rich/organic particles, $^{35}Cl^-$ + $^{37}Cl^-$ (m/z = 35 + 37) for sea salts and $Cl^-$-rich mineral dust, and $CN^-$ + $CNO^-$ + $NO_2^-$ (m/z = 28 + 42 +48) for soluble organics. The peak areas of all four ionic species were

calculated in particulates analyzed in an initial sample run of DS14-06, considered the control experiment. The sample was twice filtered using a 0.02-$\mu$m sterile mesh filter to remove insoluble particles. The sample was then doped with 746 nm PSL particles to act as a carrier of any soluble material. A blank test was performed using DI water doped with the same 746 nm PSL particles.

For all four species the observed peak areas decreased for the filtered (insoluble particle-free)
experiments to the DI and filtered particle tests (Figure 5). Wilcoxon rank sum tests were used to determine the difference in the median peak area of the two experiments (i.e., the DS14-06 sample run versus the DI and the PSL-doped DS14-06 sample run). In both the ($PO_2^-$ + $PO_3$) and ($CN^-$ + $CNO^-$ + $NO_2^-$) tests the two experiments' median peak areas were significantly different ($p < 0.05$), indicating that negligible post-aqueous coating of soluble phosphorous ($PO_2^-$ + $PO_3^-$) or soluble organics ($CN^-$ +
$CNO^-$ + $NO_2^-$) were found on the PSLs. These results suggest that P-rich particulates (the biological and P-rich inorganic classes) and organo-nitrogen particulates measured in PALMS are likely comprised predominantly of the original insoluble particle. This was not the case for the chlorine and sulfate-containing particles, indicating these materials were found in the solution. Since sea salt and sulfate-rich/organic particles are soluble and a significant source of atmospheric chlorine and sulfate (Murphy
et al., 1998), this suggests that when using this or similar nebulization methodology, these particle classes (1) cannot be directly retrieved, and (2) cause coating of insoluble particles. More generally, the relatively small particle sizes of the organic/sulfur-rich particles at both GW14 and DS14 (Fig. 4) suggests that post-aqueous coating could be increasingly important for smaller sized particles.

### 3.2.4 Feasibility of PALMS for particle concentration determinations

To incorporate PALMS as a quantitative method for discerning mass concentration within ice requires more particles than were measured during the 10 minute sample runs. To evaluate the feasibility of this method, sample DS14-05 was ran for an hour in both the positive and negative ion





modes. This test resulted in multiple classes containing >500 particles, the threshold number of particles employed here for class-dependent mass concentration measurements, but still an order of magnitude smaller than the number suggested by Schwarz et al. (2012) for laser-induced incandescent methods. Note that organic/sulfur-rich particles were not considered due to the uncertainty regarding post-

aqueous processing (see last section). After the method of Murphy et al. (2008) and Schwarz et al. (2012), representative class densities were taken as 0.8 g cm$^{-3}$ for soot, 2.7 g cm$^{-3}$ for mineral dust/metallic particles, and 1.3 g cm$^{-3}$ for biomass burning and biological particles. To calculate mass concentrations, particle size distributions for each class were integrated over the order of magnitude size interval $D_p$ = 0.25 nm to 2.5 µm using eq. (4) and parameterized using the experimentally derived

extraction efficiency provided in eq. (5). The mass concentration results are shown in Table 3. The mineral/metallic particle class (including the Ca-rich and processed particles) had the highest concentration of particles (consistent with the high relative abundance in DS14-05), estimated at 10.1 $\pm$ 5.7 ng g$^{-1}$, while the biomass-burning/biological, P-rich biological, and the soot particle classes were estimated at 5.6 $\pm$ 2.5, 7.0 $\pm$ 2.6, and 1.6 $\pm$ 0.7 ng g$^{-1}$, respectively.

15       Direct comparison of these concentrations to previously published values is complicated by differences in the particle size range retrieved by various analytical techniques as well as site-to-site differences. For example, Coulter counter techniques can extend the size-range of insoluble particle retrieval up to 50 µm (Delmonte et al. 2002) where a few particles near the upper limit would dominate the mass. Where comparisons are possible, the concentrations reported here are in line with what has

been previously reported. Mineral/metallic concentrations of 10.1 (± 5.7) ng g$^{-1}$ for DS14-05 are within an order-of-magnitude of that reported by Bory et al. (2002) for recent snowfall at the higher-elevation, inland NGRIP site in Greenland. A more direct comparison of methodologies is available from black carbon measurements made recently on DS15, a firn core collected in 2015 adjacent to the DS14 site. Using the SP2 method (following the methodology of McConnell et al., 2007; see also Wendl et al.,

2014), a mean black carbon concentration in the upper 5 m of DS15 was found to be 0.78 $\pm$ 0.97 ppb (J. McConnell, *pers. comm.*), within range of the soot concentrations (1.6 $\pm$ 0.7 ng g$^{-1}$) calculated using PALMS on the selected core segment of DS14.





### 4. Conclusions

In this study we develop and apply a new methodology utilizing SPMS to characterize particulates trapped in snow and ice at the single particle level. We show that a single instrument, in this case PALMS, can be used to discern the aerodynamic size, composition, and concentration of
insoluble ice core particulates. This online method reduces preparation time and resources required for filter-based particle retrieval methods. Based on compositional differences in particles found in two Greenlandic firn cores, we define 8 distinct particle classes in the positive ion mode and 6 in the negative ion mode (Tables 1 and 2; Appendix 1). These classes are a combination of common atmospheric types previously described in the literature and types relatively rare in the atmosphere but
common in the ice samples. Differences in the relative abundances of classes found between the two sites are consistent with their unique geography and climatology, notably, marine versus high altitude/inland locations. Furthermore, this study demonstrates that PALMS can be used to infer a sample's mass concentration using an external calibration of a size-dependent extraction efficiency which parameterizes the nebulization, delivery, and ablation/ionization of particles from concentration
slurries (Wendl et al., 2014).

We find that two classes, organic/sulfate and sea-salt particulates, appear to be artifacts of post-aqueous processing, highlighting the need for further evaluation of these particle classes in future studies. More broadly, this phenomenon is a critical feature of particles analyzed from a liquid suspension, which has not been fully appreciated in previous studies (e.g. Ault et al., 2011; Creamean et
al., 2014): completely soluble aerosols or soluble fractions of particles are homogenized into the melt solution from which insoluble components are extracted. These components are then present as small, solute-only, particles and as a surface coating on insoluble particles. The former were typically smaller than the PALMS threshold size but this need not be the case in future studies using different instruments. While the concentration of soluble material can be reduced by dilution, at this time there is
no solution to quantifying the original soluble aerosol, finding the original association of soluble with insoluble components, or completely removing this material from the analysis step.

We group the future application of SPMS to ice core studies into qualitative and quantitative uses. We show that SPMS can readily be used for qualitative analysis, as is done in atmospheric studies





(Murphy, 2005); future application may include particle classification and source-allocation using mass-spectral compositional markers. This may be of particular use in climatic transitions occurring deep in the ice core record, where thinning of the ice column requires increasingly smaller sample consumption for high-resolution measurements. Quantitatively, we show that mass concentration is feasible but with

5    long sample periods required for statistically-relevant conclusions at current instrument rates. Measurement frequency during an hour-long period registered ~1-2 particle $sec^{-1}$. Instrumental improvements, e.g. sample concentrations, leading to ~10 $sec^{-1}$ could reduce the required sample by a similar factor. This suggests that further instrument improvements in extraction efficiency, sample concentration, data rate and laser repetition rate would correspondingly reduce sampling time.

**Data Availability**

All data used in the generation of this manuscript is archived at MIT and the Harvard Dataverse and is available upon request.

15   **Appendix**

In this section, we provide exemplary particle spectra for each class, including the traditional particulate classes using the algorithm described in Cziczo et al. (2013), as well as the additional particle classes used in this study, in both the positive and negative ion mode.





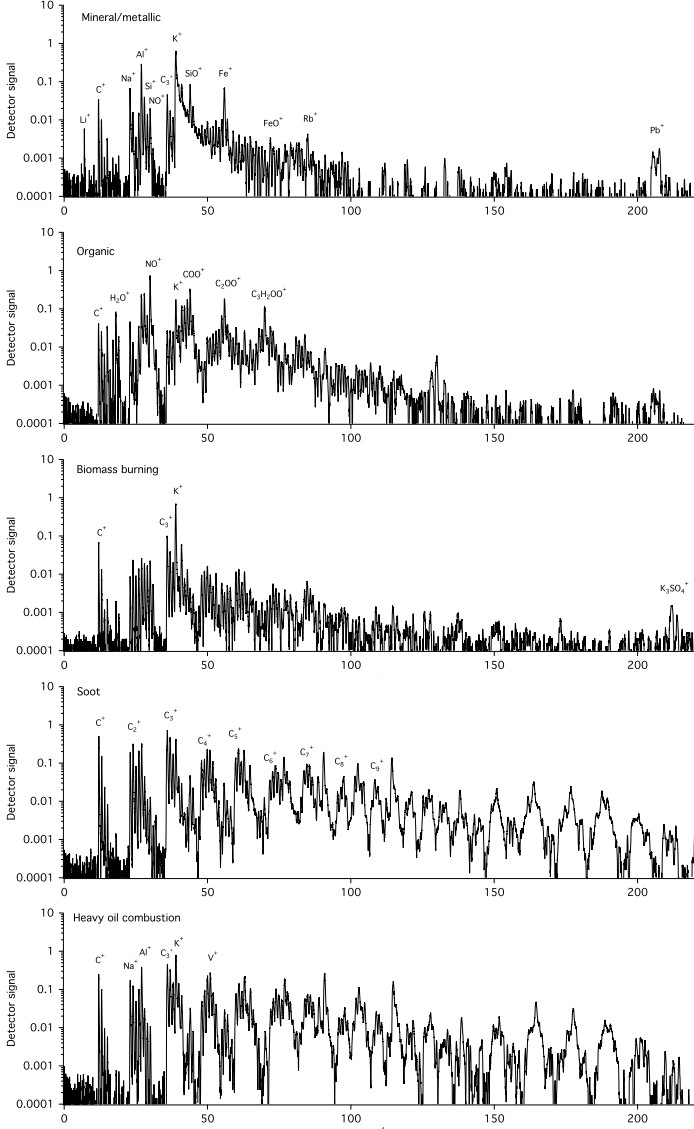

Fig. A1. Exemplary spectra taken from the DS14 and GW14 samples for the traditional positive ion mode particle classes using the classification algorithm described in Cziczo et al. (2013). From top to bottom: mineral/metallic, organic, biomass burning, soot, and heavy oil combustion/vanadium-rich.





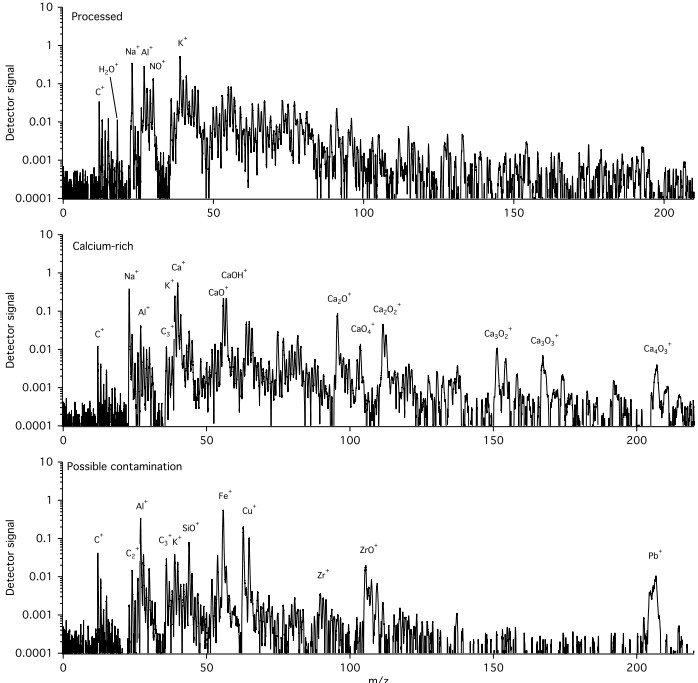

Figure A2. Exemplary spectra taken from the DS14 and GW14 samples for three positive ion mode classes commonly observed in the ice core data. From top to bottom: processed, Ca-rich, and contamination/heavy-metal. These classes were originally categorized in the mineral/metallic positive ion category using the algorithm of Cziczo et al., (2013).





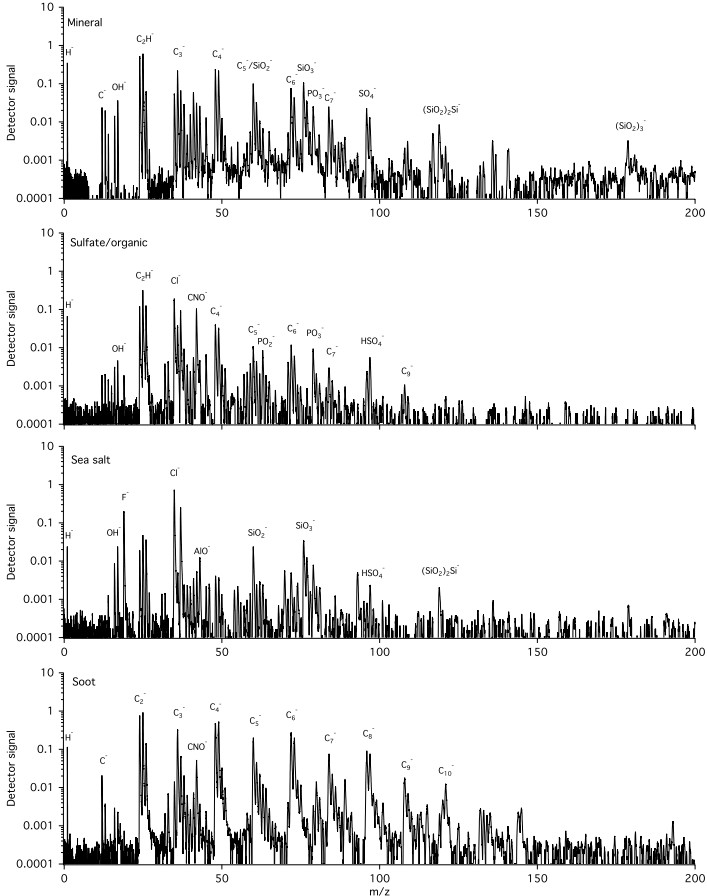

Fig. A3. Exemplary spectra taken from the DS14 and GW14 samples for the traditional negative ion mode particle classes using the classification algorithm described in Cziczo et al. (2013). From top to bottom: mineral, sulfate/organic, sea salt, and soot.





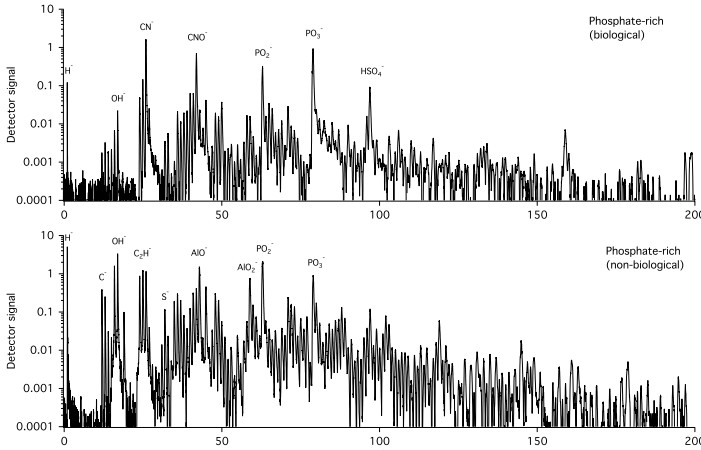

Figure A4. Exemplary spectra taken from the DS14 and GW14 samples for two negative ion mode classes commonly observed in the ice core data. From top to bottom: biological and P-rich inorganic. Both classes were originally categorized in the sulfate/organic negative-ion category using the algorithm of Cziczo et al., (2013).

**Author Contribution**

M.O. helped design the experiment, prepared the ice core samples, analyzed the samples, contributed to the data analyses and interpretation of results, and wrote the manuscript. M.Z. helped design the experiment, and contributed to the data analyses and interpretation of results. S.B.D. collected the DS14 and GW14 ice cores, assisted with ice core sample preparation, and contributed to the interpretation of results. D.J.C. helped design the experiment, supervised the laboratory work, and contributed to the data analyses and interpretation of results. All authors participated in writing and editing the manuscript.

**Competing interests**

The authors declare that they have no competing conflicts of interest.

**Acknowledgements**

This work was supported by an internal Reed Grant from MIT and National Science Foundation award



PLR-1205196 to S.B.D. M.O. acknowledges government support awarded by DoD, Air Force Office of Scientific Research, National Defense Science and Engineering Graduate (NDSEG) Fellowship, 32 CFR 168a. M.A.Z. acknowledges the support of NASA Earth and Space Science Fellowship. D.J.C. acknowledges the support of the Victor P. Starr Career Development Chair at MIT.



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





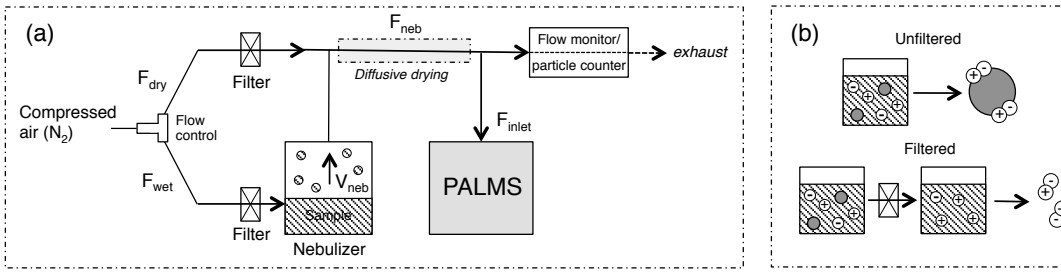

Figure 1. (a) Schematic of the experimental setup. Inert air ($N_2$) is split into dry and wet flows, with the latter, $V_{neb}$, sent to nebulize the melted ice. The low relative humidity of the combined flow evaporates condensed water and sends dried particles to PALMS. Air and liquid flows (F and V) were continuously monitored. Parker IDN-4G filters were used to remove particles from the dry nitrogen gas. (b) Idealized representation of the two types of particles produced by a slurry of insoluble particles in a solution. In the top case an insoluble particle is coated with soluble ionic material initially dissolved in the condensed liquid. In the bottom case an insoluble particle-free solution produces residual particles of the dissolved material (see text for details).





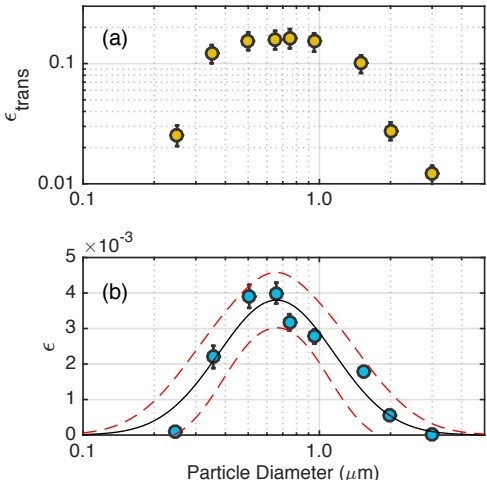

Figure 2. (a) PALMS transmission efficiency, $\boldsymbol{\epsilon_{trans}}$, as a function of aerodynamic particle size. (b) The system extraction efficiency, $\boldsymbol{\epsilon}$, as a function of particle size. The black line is the least squares lognormal regression with 95% confidence limits as red-dashed lines. Error bars represent the standard propagated error for all parameters in eq. (1).



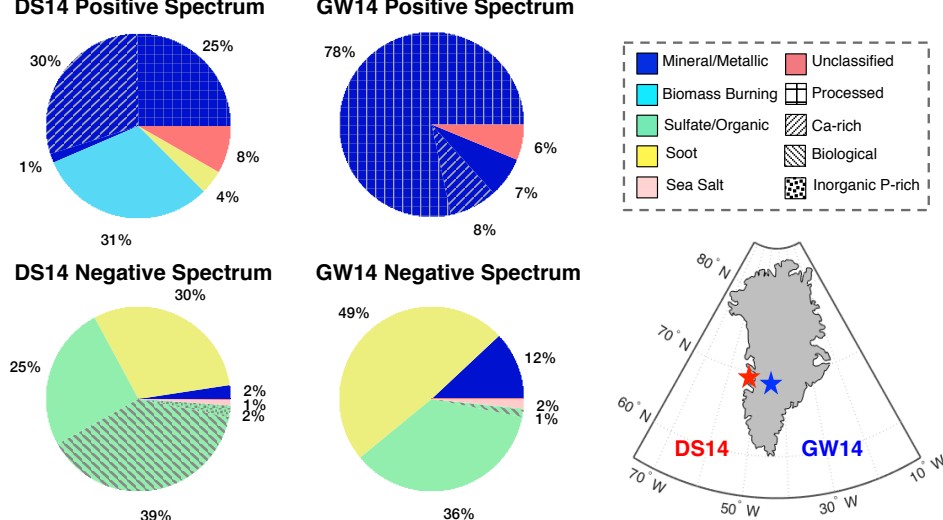

Figure 3. Particle class abundances at the DS14 and GW14 sites, and their relative locations in Greenland (lower left). The Ca-rich and processed particle sub-classes discussed in the text are associated here with the mineral/metallic classes. Likewise, and biological and inorganic P-rich sub-classes are associated with the sulfate/organic class. Particle classes below the 1% abundance level are not shown and contaminant particles from ice core processing were eliminated before data analysis. See Sect. 3.2 for additional details.





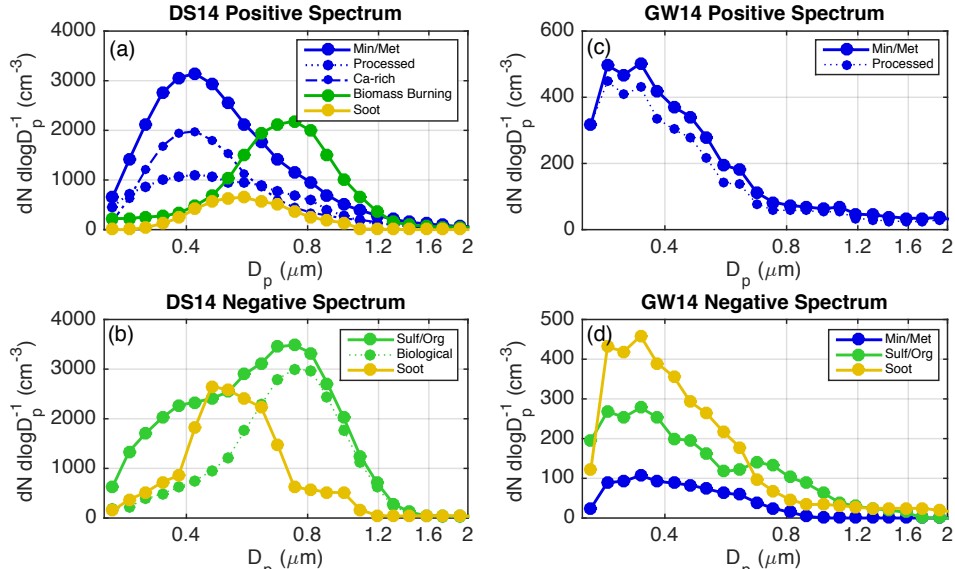

Figure 4: Log-normalized 3-pt smoothed size distributions for particle classes containing >100 particles at DS14 (a-b) and GW14 (c-d) in positive (a-c) and negative (b-d) ion mode. Due to the PALMS lower size limit the x-axis origin is 0.25 μm.





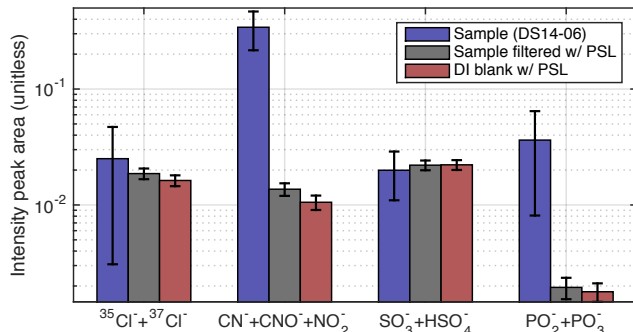

Figure 5. Average normalized peak intensity for $PO_2^- + PO_3^-$ (m/z = 63 + 79), $SO_3^- + HSO_4^-$ (m/z = 80 + 97), $^{35}Cl^- + ^{37}Cl^-$ (m/z = 35 + 37), and $CN^- + CNO^- + NO_2^-$ (m/z = 28 + 42 +48) for particles in DS14-06 filtering experiments: unfiltered (control; blue), filter doped with PSL particles (grey), and DI-water blank doped with PSL particles (red). Error bars represent one median absolute deviation about the median, estimated using a bootstrap (resampling) approach ($n$ = 1000). Note the post-aqueous coating in these four quantities exhibits no significant difference between the DI blank and the filtered samples.





**Tables**

Table 1. Positive spectra classification.

| Particle Class | Primary Identifier |
|---|---|
| Mineral/metallic | $Na^+$, $Al^+$, $K^+$, $Fe^+$ present, possible $Si^+$ and/or $SiO^+$ |
| Calcium-rich | $Ca^+$, $CaO^+$, $CaOH^+$ and $Ca_2O^+$ and/or $CaKO^+$ present |
| Processed | Combination of $Na^+$, $Al^+$, $K^+$, $Fe^+$ and abundant organics, sulfates and nitrates |
| Organic | Organics, $NO^+$, $NO_2^+$, no other characteristic markers |
| Biomass burning/biological | High $K^+$, sulfates and organics without other mineral/metallic markers |
| Soot | $C^+$, $C_2^+$, $C_3^+$, $C_4^+$, etc. V+ and VO+ |
| Sea salt | $Na^+$, $K^+$, $Na_2Cl^+$ |
| Heavy oil combustion | $V^+$ and $VO^+$ |
| Heavy metal (contamination) | High $Fe^+$, possibly $Al^+$, $Mo^+$ and/or $Sn^+$ without other mineral/metallic markers (stainless steel) $Ti^+$ and $TiO^+$ without other mineral/metallic markers (processing saw) $Zr^+$ and $ZrO^+$ (processing saw) |

Table 2. Negative spectra classification.

| Particle Class | Primary Identifier |
|---|---|
| Biological[1] | $PO_2^-$, $PO_3^-$, $CN^-$, $CNO^-$ |
| P-rich (inorganic)[1] | $PO_2^-$, $PO_3^-$, $CN^-$, $CNO^-$ |
| Mineral | $SiO_2^-$, $SiO_3^-$, possible $SiO_2AlO_2^-$ |
| Sea salt | $F^-$, $Cl^-$ no other characteristic markers |
| Soot | $C^-$, $C_2^-$, $C_3^-$, $C_4^-$, etc. |
| Sulfate/organic | $CN^-$, $CNO^-$, other organics, $HSO_4^-$ no other characteristic markers |

[1]Classes differentiated by ratios of $PO_3^-$ to $PO_2^-$ and $CN^-$ to $CNO^-$, as described in Zawadowicz et al. (2016).

10  Table 3. DS14-05 particle relative abundance (RA) and calculated mass concentration values. The values of $\rho$ are inferred, based on prior studies (see Sect. 3.2.3).

| Particle Class | Particle count (RA,%) | $\rho$ (g cm$^{-3}$) | Concentration (ng g$^{-1}$) ($\pm 1\sigma$) |
|---|---|---|---|
| Positive spectrum | | | |
| Mineral/metallic[1] | 1229 (54.8%) | 2.7 | 10.1 (±5.7) |
| Biomass burning/biological | 721 (32.1%) | 1.3 | 5.6 (±2.5) |
| Negative Spectrum | | | |
| P-rich (biological) | 970 (40.3%) | 1.3 | 7.0 (±2.6) |
| Soot | 706 (29.3%) | 0.8 | 1.6 (±0.7) |

[1]Includes pure mineral/metallic particles, as well as the subsidiary Ca-rich and Processed particle classes.