# Peer review of "Real time analysis of insoluble particles in glacial ice using single particle mass spectrometry"

_Atmospheric Measurement Techniques, 2017_

## Referee Comment (RC1) · Anonymous Referee #2 · 19 Jul 2017

\*\*\*\* Please see attached PDF, as I could not copy equations into this section \*\*\*\*

Referee Comments for Osman, M. et al., "Real time analysis of insoluble particles in glacial ice using single particle mass spectrometry"

General Comments

The authors implement a PALMS to successfully employ nebulizer+instrument techniques previously used by single particle soot photometers to make measurements of particle classification, size and concentration in ice core samples from Greenland. Though limited by a low particle transmission that is currently inherent to the PALMS instrument, the methodology still manages to result in new, interesting mea-

surements that to my knowledge have not been realized before, making this a very worthy manuscript for publication after addressing some relatively minor concerns. The results also provide substantial motivation to continue to push the transmission-efficiency capabilities of the PALMS instrument, the results of which would make the methods used here a much more viable path towards measurement of particle concentrations and size distributions in ice cores.

The presentation of the goals, setup, methodology and results are generally very clearly stated, with very few exceptions (notes below). Further, the manuscript seems to have been carefully prepared, as I struggled to find any typos, spelling errors or poor grammar. Below, I suggest some minor corrections to be addressed, including explanation of their monitoring of potential background contamination levels and system stability. Also, I suspect there may be an error in their calculation of nebulization efficiency (though it results in only a small change in the quoted number). Finally, I recommend more carefully explaining the differences in interpreting PALMS vs SP2 measurements of black carbon / soot.

Specific Comments

Page 3, line 18: Also see Katich et al., 2017 (doi:10.1080/02786826.2017.1280597), which provides a lengthy closely-related discussion on aerosolizing particulate from snow and ice.

Page 7, line 5,6: Did the authors intersperse regular measurements of 'blanks' (i.e. ultra-pure water) to quantify the average background level of particulate seen by PALMS when using a 'clean' nebulization system? There is mention of looking for background from the stainless steel band saw (not what I'm concerned about here) and of sonicating the parts between samples. But I wonder if there is a quantification of average background levels due to any residual particulate in the nebulizer lines? How does this compare to signal levels? Negligible?

Page 8, line 19: Please clarify the phrase "rate of liquid nebulization" . . . same as rate

at which liquid is fed to the nebulizer i.e. liquid uptake rate? Does this occur at a user-controlled pump rate, or is it self-aspirating? If you have control over the pump rate, this could be another way to tweak the rate of particulate delivered to the PALMS inlet.

Page 9, line 23: Agreed, the nebulization efficiency can drift over time, even substantially, depending on the solution being nebulized. Was this monitored by occasionally measuring transmission of the 8.8e6 particles/cc solution in between samples? If so, perhaps show a summary of nebulization stability in supplemental material?

Page 11, line 6: The statement, "Scaling the efficiency curve by the ratio of excess-flow to the PALMS inlet flow", I believe should read "Scaling the efficiency curve by the ratio of TOTAL-flow to the PALMS inlet flow."

Further, I don't think I agree with the calculation of the nebulization efficiency, where it is achieved simply by making the correction due to particle loss from low PALMS sample flow. I would argue the following: If a nebulizer's efficiency is defined as the ratio of 'the rate of particles emerging in aerosol from the nebulizer' (call it Raerosol = Naerosol/sec) to 'the rate of particles introduced to the nebulizer' (call it Rintroduced, which is known from your known PSL concentration and liquid uptake rate), then to know Naerosol, you have to work backwards from the number of particles that PALMS sees.

****THIS DID NOT COPY / PASTE PROPERLY. PLEASE SEE ATTACHED PDF FOR EQUATIONS****

working backward from the number of particles need by PALMS. . .

(1) (2) (3) (4) where (1) = fPALMS (2) = ïĄětrans (3) = the correction introduced in the text (4) = needed to correct for the fact that only 2/5 of the flow into PALMS has gone through the nebulizer (i.e. Fdry is just dilution air)

thus. . .

Plugging in some maximal numbers from figure 2, I get:

it's not far off the quoted 4%, but as a matter of correctness, should be changed (assuming I've not erred).

Page 16, line 6: Regarding the 0.8 g/cm3 density used for soot...What density are you referring to? Void free rBC? All recent literature that I know of uses either 2.0 g/cm3 (a bit of an old number) or 1.8 g/cm3, so maybe this is a typo, or maybe you have actually used the wrong number, or maybe you are referring to a density that isn't clarified. Please comment...

Figure 4: It is important to note somewhere that the soot size distributions here are not directly comparable to typical rBC size distributions shown in literature that are measured via incandescence, i.e. SP2 measurements. My understanding is that PALMS will only measure the size of the entire soot-containing particle (via a scattered-light signal), which includes any 'coating' that is combined with the BC particle, and is not a measurement of the 'core' refractory BC mass. On the other hand, SP2 measurements will separate the rBC core mass (or volume-equivalent-diameter, VED) from the coating associated with an individual rBC particle. This rBC core VED distribution is what is typically shown in literature. So one could not compare the soot distributions shown here to, say, Schwarz et al. 2013 ("Black Carbon Aerosol Size in Snow"). A slightly expanded discussion on the interpretation of SP2 vs PALMS measurements of soot/rBC is recommended.

Technical Corrections

Page 3, line 15: 'Schwartz' should be Schwarz.

Please also note the supplement to this comment:
https://www.atmos-meas-tech-discuss.net/amt-2017-151/amt-2017-151-RC1-supplement.pdf

**Supplement:**

**Referee Comments for Osman, M. et al.,**

**"Real time analysis of insoluble particles in glacial ice using single particle mass spectrometry"**

**General Comments**

The authors implement a PALMS to successfully employ nebulizer+instrument techniques previously used by single particle soot photometers to make measurements of particle classification, size and concentration in ice core samples from Greenland. Though limited by a low particle transmission that is currently inherent to the PALMS instrument, the methodology still manages to result in new, interesting measurements that to my knowledge have not been realized before, making this a very worthy manuscript for publication after addressing some relatively minor concerns. The results also provide substantial motivation to continue to push the transmission-efficiency capabilities of the PALMS instrument, the results of which would make the methods used here a much more viable path towards measurement of particle concentrations and size distributions in ice cores.

The presentation of the goals, setup, methodology and results are generally very clearly stated, with very few exceptions (notes below). Further, the manuscript seems to have been carefully prepared, as I struggled to find any typos, spelling errors or poor grammar. Below, I suggest some minor corrections to be addressed, including explanation of their monitoring of potential background contamination levels and system stability. Also, I suspect there may be an error in their calculation of nebulization efficiency (though it results in only a small change in the quoted number). Finally, I recommend more carefully explaining the differences in interpreting PALMS vs SP2 measurements of black carbon / soot.

**Specific Comments**

**Page 3, line 18**:  Also see Katich et al., 2017 (doi:10.1080/02786826.2017.1280597), which provides a lengthy closely-related discussion on aerosolizing particulate from snow and ice.

**Page 7, line 5,6**:  Did the authors intersperse regular measurements of 'blanks' (i.e. ultra-pure water) to quantify the average background level of particulate seen by PALMS when using a 'clean' nebulization system? There is mention of looking for background from the stainless steel band saw (not what I'm concerned about here) and of sonicating the parts between samples. But I wonder if there is a quantification of average background levels due to any residual particulate in the nebulizer lines?  How does this compare to signal levels? Negligible?

**Page 8, line 19**:  Please clarify the phrase "rate of liquid nebulization" … same as rate at which liquid is fed to the nebulizer i.e. liquid uptake rate?  Does this occur at a user-controlled pump rate, or is it self-aspirating? If you have control over the pump rate, this could be another way to tweak the rate of particulate delivered to the PALMS inlet.

**Page 9, line 23**:  Agreed, the nebulization efficiency can drift over time, even substantially, depending on the solution being nebulized. Was this monitored by occasionally measuring transmission of the 8.8e6 particles/cc solution in between samples?  If so, perhaps show a summary of nebulization stability in supplemental material?

**Page 11, line 6**:  The statement, "Scaling the efficiency curve by the ratio of excess-flow to the PALMS inlet flow", I believe should read "Scaling the efficiency curve by the ratio of **TOTAL**-flow to the PALMS inlet flow."

Further, I don't think I agree with the calculation of the nebulization efficiency, where it is achieved simply by making the correction due to particle loss from low PALMS sample flow. I would argue the following:  If a nebulizer's efficiency is defined as the ratio of 'the rate of particles emerging in aerosol from the nebulizer' (call it $R_{aerosol} = N_{aerosol}/sec$) to 'the rate of particles introduced to the nebulizer' (call it $R_{introduced}$, which is known from your known PSL concentration and liquid uptake rate), then to know $N_{aerosol}$, you have to work backwards from the number of particles that PALMS sees.

$$\varepsilon_{\text{nebulization}} = \frac{R_{aerosol}}{R_{introduced}} = \frac{N_{aerosol}/sec}{R_{introduced}} = \frac{N_{aerosol}/sec}{m_{psl} \cdot V_{neb}}$$

working backward from the number of particles need by PALMS…

$$\left(N_{aerosol}/sec\right) = \underbrace{\left(\frac{N_{particles}\,detected\,by\,PALMS}{sec}\right)}_{(1)} \cdot \underbrace{\left(\frac{N_{particles}\,introduced\,to\,PALMS}{N_{particles}\,detected\,by\,PALMS}\right)}_{(2)} \cdot \underbrace{\left(\frac{F_{wet}+F_{dry}}{F_{inlet}}\right)}_{(3)}\underbrace{\left(\frac{F_{flow}}{F_{dry}}\right)}_{(4)}$$

where
(1) = $f_{PALMS}$
(2) = $\varepsilon_{trans}$
(3) = the correction introduced in the text
(4) = needed to correct for the fact that only 2/5 of the flow into PALMS has gone through the nebulizer (i.e. $F_{dry}$ is just dilution air)

thus…

$$\varepsilon_{neb} = \varepsilon_{PSL} \cdot \frac{1}{\varepsilon_{trans}} \cdot \left(\frac{F_{wet}+F_{dry}}{F_{Dry}}\right) = \frac{\varepsilon_{PSL}}{\varepsilon_{trans}} \cdot \left(\frac{F_{flow}}{F_{Dry}}\right)$$

Plugging in some maximal numbers from figure 2, I get:

$$\varepsilon_{neb(\max)} = \left(\frac{0.004}{0.15}\right) \cdot \left(\frac{5}{2}\right) \cong 6.7\%$$

it's not far off the quoted 4%, but as a matter of correctness, should be changed (assuming I've not erred).

**Page 16, line 6:** Regarding the 0.8 g/cm$^3$ density used for soot…What density are you referring to? Void free rBC? All recent literature that I know of uses either 2.0 g/cm$^3$ (a bit of an old number) or 1.8 g/cm$^3$, so maybe this is a typo, or maybe you have actually used the wrong number, or maybe you are referring to a density that isn't clarified. Please comment…

**Figure 4:** It is important to note somewhere that the soot size distributions here are not directly comparable to typical rBC size distributions shown in literature that are measured via incandescence, i.e. SP2 measurements. My understanding is that PALMS will only measure the size of the entire soot-containing particle (via a scattered-light signal), which includes any 'coating' that is combined with the BC particle, and is not a measurement of the 'core' refractory BC mass. On the other hand, SP2 measurements will separate the rBC core mass (or volume-equivalent-diameter, VED) from the coating associated with an individual rBC particle. This rBC core VED distribution is what is typically shown in literature. So one could not compare the soot distributions shown here to, say, Schwarz et al. 2013 ("Black Carbon Aerosol Size in Snow"). A slightly expanded discussion on the interpretation of SP2 vs PALMS measurements of soot/rBC is recommended.

**Technical Corrections**

Page 3, line 15: 'Schwartz' should be Schwarz.

---

## Referee Comment (RC2) · Anonymous Referee #1 · 24 Jul 2017

Review to "Real time analysis of insoluble particles in glacial ice using single particle mass spectrometry" by Osman et al., AMTD 2017

The manuscript by Osman and coworkers describes the application of a laser ablation single particle mass spectrometer on the analysis of particles in ice core samples. The authors describe the extraction of particles from the samples, the efficiency of particle transfer into the mass spectrometer and also attempt to perform a quantitative measurement of mass concentrations. The manuscript is well written and fits into the scope of AMT. Methods and results are clearly explained and presented. However, I have one major issue that is described below:

[Figure]

Major comments

While I do not have any concerns about the chemical analysis, which has been done by SPMS numerous times for atmospheric particles, I have serious problems with the mass concentration measurement:

- The size dependent extraction efficiency curve was measured using PSL particles, a method that is called "external calibration" (referenced to Wendl et al., which is not an SPMS but an SP2 paper).

- The measured efficiencies are very low (highest values for 657 nm PSL particles are around 0.40%). This results in a very large correction factor needed to derive the mass concentration in the sample

- The efficiency is assumed to be independent of shape and composition. This is certainly an oversimplification.

- Drifts in nebulizer efficiency have not been considered.

During the PSL calibration experiments, the authors used an OPS to monitor particle concentration and size to measure the transmission efficiency in to the PALMS (which was found to be between 1 and – 16%). Why has this (or another method to measure size and number) not been done during the ice core sample analysis? By such, the transmission efficiency into the PALMS would have been measured using the real sample particles. Non-spherical shapes of the insoluble particles can certainly influence the transmission efficiency.

Furthermore, methods exist to measure particle concentration in liquids. A comparison of the derived particle concentrations from the SPMS with such a reference measurement would have helped in validating the results.

At least a determination of the mass concentration by filtering the solution and weighing the filter would have been possible, although here larger particle may dominate the total mass concentration Without any comparison to an independent measurement of the

same sample, the quantification of the results seems not reliable and overstretches the capabilities of an SPMS.

Minor comments:

Page 11, lines 1-3: How was determined that all water was evaporated?

Figures of mass spectra in appendix: What is the method to select which mass peaks are labeled. Apparently not always the largest peaks? Are peaks unknown/not identified, e.g. in Fig. A1, lowest panel, around m/z 90? Additionally, minor tick marks and/or grid lines help the reader to determine the m/z number of a peak in the mass spectrum.

---

## Author Comment (AC1) · 3 Oct 2017

> *Dear Anonymous Reviewer #2,*
>
> *We thank you for your valuable feedback. We have reviewed all your comments/suggestions, and have attempted to address each to the best of our ability below. Please refer to italicized and indented portions for responses.*

Referee Comments for Osman, M. et al., "Real time analysis of insoluble particles in glacial ice using single particle mass spectrometry"

**General Comments**

The authors implement a PALMS to successfully employ nebulizer+instrument techniques previously used by single particle soot photometers to make measurements of particle classification, size and concentration in ice core samples from Greenland. Though limited by a low particle transmission that is currently inherent to the PALMS instrument, the methodology still manages to result in new, interesting measurements that to my knowledge have not been realized before, making this a very worthy manuscript for publication after addressing some relatively minor concerns.

The results also provide substantial motivation to continue to push the transmission efficiency capabilities of the PALMS instrument, the results of which would make the methods used here a much more viable path towards measurement of particle concentrations and size distributions in ice cores. The presentation of the goals, setup, methodology and results are generally very clearly stated, with very few exceptions (notes below). Further, the manuscript seems to have been carefully prepared, as I struggled to find any typos, spelling errors or poor grammar. Below, I suggest some minor corrections to be addressed, including explanation of their monitoring of potential background contamination levels and system stability. Also, I suspect there may be an error in their calculation of nebulization efficiency (though it results in only a small change in the quoted number). Finally, I recommend more carefully explaining the differences in interpreting PALMS vs SP2 measurements of black carbon / soot.

**Specific Comments**

C1) Page 3, line 18: Also see Katich et al., 2017 (doi:10.1080/02786826.2017.1280597), which provides a lengthy closely-related discussion on aerosolizing particulate from snow and ice.

> *We thank you for pointing us to this recent study + discussion therein, which had been previously unknown to the authors; the reference for Katich et al., 2017 has now been included (Pg. 3 Line 19).*

C2) Page 7, line 5,6: Did the authors intersperse regular measurements of 'blanks' (i.e. ultra-pure water) to quantify the average background level of particulate seen by PALMS when using a 'clean' nebulization system? There is mention of looking for background from the stainless steel band saw (not what I'm concerned about here) and of sonicating the parts between samples. But I wonder if there is a quantification of average background levels due to any residual particulate in the nebulizer lines? How does this compare to signal levels? Negligible?

*We did not systematically implement measurements of blanks between samples as the reviewer describes here. Our methodology was to flush the flow line continuously with an inert $N_2$ gas-flow (for ~15 min) between samples to remove residual particulates prior to the next measurement. To answer this point we now state the following in Sect. 2.3 (Pg. 7 Lines 8-9 ):*

*"Between runs, the nebulizer and sampling beaker were cleaned and sonicated for 15 minutes using ultrapure (Milli-Q; 18.2 MΩ) water, and the flow-line flushed continuously with the inert carrier gas."*

*We do acknowledge the utility of the reviewer's suggestion to regularly implement blanks between measurements and have also included the following in 3.2.4 (Pg. 17-18 Lines 25-29, 1-7):*

*"While our results show potential exists for using SPMS to determine insoluble mass concentrations of particles in snow and ice, they also identify areas where more work is needed before SPMS can be used as a quantitative tool. These include: i) executing multiple extraction efficiency (eq. 1) calculations as a function of particle class (in addition to size), ii) incorporating regularized tests for drifts in SPMS extraction efficiency and employing "blank" tests between sample measurements in order to improve delineation to changes in background particulate levels, iii) achieving a greater number of particle measurements (either through improvements in particle extraction/PALMS transmission or longer sample integration times), and iv) comparing SPMS-derived particle concentrations with results from alternate, well-founded high-precision instrumentation (e.g., an Ultra-High Sensitivity Aerosol Spectrometer (UHSAS; Droplet Measurement Technologies Inc., Boulder, CO), or Coulter Counter instrumentation)."*

C3) Page 8, line 19: Please clarify the phrase "rate of liquid nebulization"… same as rate at which liquid is fed to the nebulizer i.e. liquid uptake rate? Does this occur at a user controlled pump rate, or is it self-aspirating? If you have control over the pump rate, this could be another way to tweak the rate of particulate delivered to the PALMS inlet.

*The "rate of liquid nebulization" is the average volumetric loss rate of the liquid sample during nebulization of the sample. This rate is indirectly user-controlled by setting the "wet" gas flow rate to the nebulizer (i.e., F_wet). However, while increasing the wet flow rate to the nebulizer could increase the rate of particulate delivered to the PALMS inlet, it was determined that water saturation (i.e., quenching) of the particles became problematic at much higher rates than that quoted in the manuscript (2 lpm). The following text has been added to Sect. 2.3.2 (Pg. 8, lines 23-24) to clarify:*

*"...$V_{neb}$ is the rate of liquid nebulization (i.e., a prescribed volumetric loss rate of the sample, determined here using a scale; $4.4 \cdot 10^{-6} \pm 1.6 \cdot 10^{-6}$ mL sec$^{-1}$)"*

C4) Page 9, line 23: Agreed, the nebulization efficiency can drift over time, even substantially, depending on the solution being nebulized. Was this monitored by occasionally measuring transmission of the 8.8e6 particles/cc solution in between samples? If so, perhaps show a summary of nebulization stability in supplemental material?

*Since only one sample (DS14-05) was measured for quantitative purposes, transmission of the 8.8e6 particles/cc solution was not tested between sample(s). We did, however, test whether systematic trends in nebulization drift could occur over the hour-long measurement period. This test was done by directing a particle-laden airflow (nebulized from the monodisperse, 746 nm PSL liquid standard: $m_{PSL}(D_p = 746$ nm) $= 8.8 \times 10^6$ PSL particles cm$^{-3}$; Sect. 2.3.3), to an optical particle sizer (OPS; MesaLabs Bios DryCal 220), and performing continuous, one-second interval measurements over three separate ~1-hour long tests (i.e., the longest sample integration period in the manuscript). The nebulization efficiency ($\varepsilon_{neb}$) was calculated in this test as,*

$$\varepsilon_{neb}\big(D_p = 746 \ nm\big) = \frac{n_{OPS}(D_p = 746 \ nm) \cdot F_{flow}}{m_{PSL}(D_p = 746 \ nm) \cdot V_{neb}}$$

*where $n_{OPS}$ is the PSL number concentration measured by the OPS, and the flow rates $F_{neb}$, $V_{neb}$ and, $F_{wet}$ are as described in the main text. The long-term drifts in nebulization, calculated as the linear percent change over the hour-long measurement interval, were determined in the three tests to be 22%, 9.2%, and -33 % ($\Delta\varepsilon_{neb}/\Delta t = 0.18 \cdot 10^{-5}$ s$^{-1}$, $0.08 \cdot 10^{-5}$ s$^{-1}$, and $-0.30 \cdot 10^{-5}$ s$^{-1}$, respectively).*

*Importantly, results of the three tests indicated that drift direction was not systematic, as both negative and positive drift biases occurred over the one-hour nebulization periods (Fig. R1, shown below). It is thus reasonable to view the drift uncertainty as a simple spread about the hour-long mean of the three tests, in this case equating to $\varepsilon_{neb}$ = 0.068 ± 0.013 (1 s.d.), or ~18% relative uncertainty.*

*It is equally important to note that in our study, calculation of particle mass-concentration (eq. 4) does not explicitly incorporate estimates of $\varepsilon_{neb}$, but rather estimates of the extraction efficiency, $\varepsilon$ (eq. 5), determined experimentally and independent of $\varepsilon_{neb}$. However, via eq. 6 (now included in the main text; see C5 below), $\varepsilon$ is shown to be a function of $\varepsilon_{neb}$ and transmission efficiency, $\varepsilon_{trans}$. Since past studies (e.g., Cziczo et al., 2006) have illustrated that PALMS transmission is relatively stable, we thus take the uncertainty interval calculated for $\varepsilon$ (~30% relative uncertainty at $\varepsilon(D_p$ = 746 nm); eq. 5) to implicitly encapsulate uncertainties in nebulization efficiency.*

*The above information has been included as supplementary material, and the following sentence added to Sect. 3.2.4 (Pg. 16 Lines 23-26):*

*"Note that while no systematic trends in nebulization drift were found over either hour-long measurement period, short term fluctuations in nebulization could occur; for the present experiment, such fluctuations are assumed to be encapsulated as uncertainty about the extraction efficiency parameterization (eq. 5; see Supplementary Material for details)."*

*Overall, we agree with the reviewer that future mass-concentration applications using SPMS – especially those where multiple successive samples are measured for mass concentration – should implement regular standardized checks for performance drift between samples. Text has also been added to 3.2.4 making this latter suggestion explicit (Pg. 17-18 Lines 25-29, 1-7; refer to C2 response).*

[Figure]

*Fig. R1. Results of three separate nebulization drift tests.*

C5) Page 11, line 6: The statement, "Scaling the efficiency curve by the ratio of excess-flow to the PALMS inlet flow", I believe should read "Scaling the efficiency curve by the ratio of TOTAL-flow to the PALMS inlet flow." Further, I don't think I agree with the calculation of the nebulization efficiency, where it is achieved simply by making the correction due to particle loss from low PALMS sample flow. I would argue the following: If a nebulizer's efficiency is defined as the ratio of 'the rate of particles emerging in aerosol from the nebulizer' (call it $R_{aerosol} = N_{aerosol}/sec$) to 'the rate of particles introduced to the nebulizer' (call it $R_{introduced}$, which is known from your known PSL concentration and liquid uptake rate), then to know $N_{aerosol}$, you have to work backwards from the number of particles that PALMS sees.

****THIS DID NOT COPY / PASTE PROPERLY. PLEASE SEE ATTACHED PDF FOR

EQUATIONS****

*For convenience, we have copy and pasted the aforementioned attached .pdf below:*

Page 11, line 6: The statement, "Scaling the efficiency curve by the ratio of excess-flow to the PALMS inlet flow", I believe should read "Scaling the efficiency curve by the ratio of TOTAL-flow to the PALMS inlet flow."

Further, I don't think I agree with the calculation of the nebulization efficiency, where it is achieved simply by making the correction due to particle loss from low PALMS sample flow. I would argue the following:  If a nebulizer's efficiency is defined as the ratio of 'the rate of particles emerging in aerosol from the nebulizer' (call it $R_{aerosol} = N_{aerosol}/sec$) to 'the rate of particles introduced to the nebulizer' (call it $R_{introduced}$, which is known from your known PSL concentration and liquid uptake rate), then to know $N_{aerosol}$, you have to work backwards from the number of particles that PALMS sees.

$$\varepsilon_{nebulization} = \frac{R_{aerosol}}{R_{introduced}} = \frac{N_{aerosol}/sec}{R_{introduced}} = \frac{N_{aerosol}/sec}{m_{psl} \cdot V_{neb}}$$

working backward from the number of particles need by PALMS...

$$\left(N_{aerosol}/sec\right) = \left(\frac{N_{particles}\,detected\;by\;PALMS}{sec}\right) \cdot \left(\frac{N_{particles}\,introduced\;to\;PALMS}{N_{particles}\,detected\;by\;PALMS}\right) \cdot \left(\frac{F_{wet}+F_{dry}}{F_{inlet}}\right)\left(\frac{F_{flow}}{F_{dry}}\right)$$

$$\qquad\qquad\qquad\qquad (1)\qquad\qquad\qquad\qquad (2)\qquad\qquad\qquad (3)\qquad (4)$$

where
(1) = $f_{PALMS}$
(2) = $\varepsilon_{trans}$
(3) = the correction introduced in the text
(4) = needed to correct for the fact that only 2/5 of the flow into PALMS has gone through the nebulizer (i.e. $F_{dry}$ is just dilution air)

thus...

$$\varepsilon_{neb} = \varepsilon_{PSL} \cdot \frac{1}{\varepsilon_{trans}} \cdot \left(\frac{F_{wet}+F_{dry}}{F_{Dry}}\right) = \frac{\varepsilon_{PSL}}{\varepsilon_{trans}} \cdot \left(\frac{F_{flow}}{F_{Dry}}\right)$$

Plugging in some maximal numbers from figure 2, I get:

$$\varepsilon_{neb(max)} = \left(\frac{0.004}{0.15}\right) \cdot \left(\frac{5}{2}\right) \cong 6.7\%$$

it's not far off the quoted 4%, but as a matter of correctness, should be changed (assuming I've not erred).

*We thank the reviewer for sharing this concern, which was well articulated by the equations provided in his/her supplement. We agree with the reviewer's determination of the nebulization efficiency (void the incorporation of F_dry, which we believe should in fact be F_wet). However, we believe the reviewer's primary underlying concern (as it relates to miscalculation of the nebulization efficiency) was due primarily to a nomenclature error: we incorrectly stated "nebulization efficiency" as opposed to "extraction efficiency" in Sect 3.1. While 1) nebulization efficiency (i.e., Raerosol/Rintroduced, as defined by the reviewer) and 2) transmission efficiency can be viewed as two independent properties of the experimental set-up, both quantities are effectively encapsulated by 3) extraction efficiency (i.e., as per the reviewer e_psl = e_neb\*e_trans\*[F_wet/F_flow]); in this context, we believe our more-simplistic scaling correction remains valid.*

*To avoid nomenclature confusion and to render our SPMS results more directly comparable to past studies (e.g., Schwarz et al., 2012, Ohata et al., 2013, Wendl et al., 2013, Katich et al., 2017), our simplistic scaling has been removed, and a revised discussion now including nebulization efficiency has been added to Sect. 3.1 (Pg. 11 Lines 18-24), including the addition of a similar derivation to that*

*of the reviewer's to Appendix 1 (Pg., 19-20, Lines, 22-27, 1-10):*

*"A1. Calculating nebulization efficiency*

*Here, we derive the determination of nebulization efficiency ($\varepsilon_{neb}$; Sect. 3.1). We define $\varepsilon_{neb}$ as the flow-weighted ratio of the rate of successfully nebulized particles per unit time relative to the (liquid) number concentration of particles introduced to the nebulizer, such that*

$$\varepsilon_{neb} = \frac{f_{neb}(D_p)}{m_{PSL}(D_p) \cdot F_{neb}} \qquad (A1)$$

*where $m_{PSL}(D_p)$ and $F_{neb}$ are as defined in eq. (1), and $f_{neb}(D_p)$ is the frequency of particles successfully nebulized (e.g., particles sec$^{-1}$) as a function of PSL diameter ($D_p$). In this case, $f_{neb}(D_p)$ is the quantity that must be solved for. We take*

$$f_{neb}(D_p) = \frac{n_{PALMS}(D_p) \cdot F_{flow}}{\varepsilon_{trans}(D_p)} \cdot \left[\frac{F_{flow}}{F_{wet}}\right] \qquad (A2)$$

*such that the scalar quantity $\left[\frac{F_{flow}}{F_{wet}}\right]$ acts as a correction for the flow balance of particles actually passing through the nebulizer (Figure 1) and $\varepsilon_{trans}(D_p)$ corrects for the size-dependent particle transmission of PALMS. Note that $n_{PALMS}(D_p)$ and $F_{flow}$ are as previously defined in eq. (1). Plugging eq. (A2) into (A1), and via relation to eq. (1),*

$$\varepsilon_{neb} = \frac{\varepsilon(D_p)}{\varepsilon_{trans}(D_p)} \cdot \left[\frac{F_{flow}}{F_{wet}}\right] \qquad (A3)$$

*as defined in eq. (6)."*

C6) Page 16, line 6: Regarding the 0.8 g/cm3 density used for soot… What density are you referring to? Void free rBC? All recent literature that I know of uses either 2.0 g/cm3 (a bit of an old number) or 1.8 g/cm3, so maybe this is a typo, or maybe you have actually used the wrong number, or maybe you are referring to a density that isn't clarified. Please comment…

*Assuming general compositional similarity between BC and soot, we used an effective soot density value of 0.8 g/cm$^3$ following the experimental work of Moteki and Kondo (2010) and Kiselev et al. (2010), and as implemented by Schwarz et al. (2012). We have therefore retained 0.8 g/cm$^3$, though have clarified that this is indeed an effective density value (Pg. 16, line 18-19).*

C7) Figure 4: It is important to note somewhere that the soot size distributions here are not directly comparable to typical rBC size distributions shown in literature that are measured via incandescence, i.e. SP2 measurements. My understanding is that PALMS will only measure the size of the entire soot-containing particle (via a scattered-light signal), which includes any 'coating' that is combined with the BC particle, and is not a measurement of the 'core' refractory BC mass. On the other hand, SP2 measurements will separate the rBC core mass (or volume-equivalent-diameter, VED) from the coating associated with an individual rBC particle. This rBC core VED distribution is what is typically shown in literature. So one could not compare the soot distributions shown here to, say, Schwarz et al. 2013 ("Black Carbon Aerosol Size in Snow"). A slightly expanded discussion on the interpretation of SP2 vs PALMS measurements of soot/rBC is recommended.

*We thank the reviewer for this valuable suggestion, and the following discussion was included at the end of Sect. 3.2.4 (Pg. 17-18 Lines 13-24):*

*"There are notable inherent differences between SPMS- and SP2-derived soot size distribution determinations, however. Namely, whereas SP2 can differentiate the volume-equivalent diameter of refractory soot-components in compositionally-heterogeneous particles (Schwarz et al., 2012), our SPMS approach presumes a particle to be comprised wholly of soot if its mass-spectrum is classified as such. Thus, depending upon the morphology, internal mixing state, and ionization potential of the analyzed soot particles, SPMS may be subject to positive size distribution biases (Fig. 4)."*

**Technical Corrections**

Page 3, line 15: 'Schwartz' should be Schwarz.

*Typo has now been corrected (Pg. 3, line 15).*

***References***:

*Cziczo, D. J., Thomson, D. S., Thompson, T. L., DeMott, P. J., Murphy, D. M: Particle analysis by laser mass spectrometry (PALMS) studies of ice nuclei and other low number density particles, Int. J. Mass Spectrometry, 258, 21-29. 2006.*

*Katich, J. M., Perring, A. E., and Schwarz, J. P.: Optimized detection of particulates from liquid samples in the aerosol phase: Focus on black carbon, Aerosol Science and Technology, 51:5, 543-553, doi: 10.1080/02786826.2017.1280597, 2017.*

*Kiselev, A., Wennrich, C., Stratmann, F., Wex, H., Henning, S., Mentel, T. F., Kiendler - Scharr, A., Schneider, J., Walter, S., and Lieberwirth, I.: Morphological characterization of soot aerosol particles during LACIS Experiment in November (LExNo), J. Geophys. Res., 115, D11204, doi:10.1029/2009JD012635, 2010.*

*Ohata, S., Moteki, N., Schwarz, J., Fahey, D., and Kondo, Y.: Evaluation of a Method to Measure Black Carbon Particles Suspended in Rainwater and Snow Samples, Aerosol Sci. Technol., 47, 1073–1082, doi:10.1080/02786826.2013.824067, 2013.*

*Schwarz, J. P., Doherty, S. J., Li, F., Ruggiero, S. T., Tanner, C. E., Perring, A. E., Gao, R. S., and Fahey, D. W.: Assessing Single Particle Soot Photometer and Integrating Sphere/Integrating Sandwich Spectrophotometer measurement techniques for quantifying black carbon concentration in snow, Atmos. Meas. Tech., 5, 2581-2592, doi:10.5194/amt-5-2581-2012, 2012.*

*Wendl, I. A., Menking, J. A., Färber, R., Gysel, M., Kaspari, S. D., Laborde, M. J. G., and Schwikowski, M.: Optimized method for black carbon analysis in ice and snow using the Single Particle Soot Photometer, Atmos. Meas. Tech., 7, 2667-2681, doi:10.5194/amt-7-2667-2014, 2014.*

---

## Author Comment (AC2) · 3 Oct 2017

Review to "Real time analysis of insoluble particles in glacial ice using single particle mass spectrometry" by Osman et al., AMTD 2017

*Dear Anonymous Reviewer #1,*

*We thank you for your valuable feedback. We have reviewed all your comments/suggestions, and have attempted to address each to the best of our ability below. Please refer to italicized and indented portions for responses.*

The manuscript by Osman and coworkers describes the application of a laser ablation single particle mass spectrometer on the analysis of particles in ice core samples. The authors describe the extraction of particles from the samples, the efficiency of particle transfer into the mass spectrometer and also attempt to perform a quantitative measurement of mass concentrations. The manuscript is well written and fits into the scope of AMT. Methods and results are clearly explained and presented. However, I have one major issue that is described below:

**Major comments**

While I do not have any concerns about the chemical analysis, which has been done by SPMS numerous times for atmospheric particles, I have serious problems with the mass concentration measurement:

C1) - The size dependent extraction efficiency curve was measured using PSL particles, a method that is called "external calibration" (referenced to Wendl et al., which is not an SPMS but an SP2 paper).

*Since the SP2 remains the most common single particle, online methodology currently employed in ice core analyses, the experimental framework presented in Wendl et al. (2014) is viewed here to be a plausible surrogate for comparisons to our new SPMS-based single particle approach. Text has now been included in Sect. 2.3.4, making it explicit that the Wendl et al. (2014) "external calibration" approach is based on SP2 methods (Pg. 9 Lines 21-24):*

*"As defined by Wendl et al. (2014), an "external" calibration approach, as commonly employed in SP2-based measurements of refractory black carbon in glacial snow and ice, assumes that ε for a given monodisperse PSL standard scales with an unknown, morphologically/compositionally-heterogeneous and polydisperse, ice core sample."*

*Additionally, to provide a more direct comparison between SP2 and SPMS techniques, we have now incorporated a new section (3.1.1; Pg. 11-12 Lines 25-27, 1-23) entitled "Comparison of SPMS to SP2-based single particle methods in snow and ice".*

C2) - The measured efficiencies are very low (highest values for 657 nm PSL particles are around 0.40%). This results in a very large correction factor needed to derive the mass concentration in the sample.

*While we acknowledge this potential fallback, we point out that the relatively low extraction efficiency is largely a result of differences inherent to SP2 vs. PALMS transmission and measurement*

*capabilities, which have now been better contrasted/clarified in Section 3.1.1 (Pg. 12 Lines 12-22):*

*"More recent studies have achieved high nebulization efficiencies (~50%) up to ~2 μm using the CETAC Marin-5 pneumatic nebulizer (Teledyne CETAC Technologies, Ohaha, NE, USA; Mori et al., 2016; Katich et al., 2017). In this context, however, it is noted that achieving too high of an extraction efficiency could be disadvantageous for SPMS, should the number of particles reaching the SPMS inlet exceed that instrument's max transmission rate (~10 particles sec-1 for PALMS; Cziczo et al., 2006) where the limit is data writing and laser repetition rate. This does not affect SP2, which can more rapidly measure the incandescence of carbonaceous material passing through a continuous laser (i.e., 2-3 orders of magnitude higher), though with the cost of i) not delivering information on internal mixing state or ii) aerodynamic size (as opposed to black-carbon volume equivalent diameter). More efficient nebulization at relevant SPMS sizes, coupled to more rapid excimer lasers and data writing, would increase the data acquisition rate."*

C3) - The efficiency is assumed to be independent of shape and composition. This is certainly an oversimplification.

*We agree with the reviewer that this is indeed an important assumption underlying our methodology. We have now included text in 2.3.2 (Pg. 9 Lines 23-24) acknowledging the assumption more explicitly:*

*"... assumes that ε for a given monodisperse PSL standard scales with an unknown, morphologically and compositionally-heterogeneous and polydisperse, ice core sample."*

*We have additionally included the following text at the end of Sect. 3.2.4 encouraging future studies to explore these assumptions in greater detail (Pg. 17-18 Lines 25-29, 1-7):*

*"While our results show potential exists for using SPMS to determine insoluble mass concentrations of particles in snow and ice, they also identify areas where more work is needed before SPMS can be used as a quantitative tool. These include: i) executing multiple extraction efficiency (eq. 1) calculations as a function of particle class (in addition to size), (...)"*

C4) - Drifts in nebulizer efficiency have not been considered.

*Since only one sample (DS14-05) was measured for quantitative purposes, transmission of the 8.8e6 particles/cc solution was not tested between sample(s). We did, however, test whether systematic trends in nebulization drift could occur over the hour-long measurement period. This test was done by directing a particle-laden airflow (nebulized from the monodisperse, 746 nm PSL liquid standard: $m_{PSL}(D_p = 746 \text{ } nm) = 8.8 \times 10^6$ PSL particles cm$^{-3}$; Sect. 2.3.3), to an optical particle sizer (OPS; MesaLabs Bios DryCal 220), and performing continuous, one-second interval measurements over three separate ~1-hour long tests (i.e., the longest sample integration period in the manuscript). The nebulization efficiency ($\varepsilon_{neb}$) was calculated in this test as,*

$$\varepsilon_{neb}\big(D_p = 746 \text{ } nm\big) = \frac{n_{OPS}(D_p=746 \text{ } nm) \cdot F_{flow}}{m_{PSL}(D_p=746 \text{ } nm) \cdot V_{neb}}$$

*where $n_{OPS}$ is the PSL number concentration measured by the OPS, and the flow rates $F_{neb}$, $V_{neb}$ and, $F_{wet}$ are as described in the main text. The long-term drifts in nebulization, calculated as the linear percent change over the hour-long measurement interval, were determined in the three tests to be 22%, 9.2%, and -33 % ($\Delta\varepsilon_{neb}/\Delta t = 0.18 \cdot 10^{-5}$ s$^{-1}$, $0.08 \cdot 10^{-5}$ s$^{-1}$, and $-0.30 \cdot 10^{-5}$ s$^{-1}$, respectively). Importantly, results of the three tests indicated that long-term drift direction was not systematic, as both negative and positive drift biases occurred over the one-hour nebulization periods (Fig. R1, shown below). It is thus reasonable to view the drift uncertainty as a simple spread about the hour-long mean of the three tests, in this case equating to $\varepsilon_{neb} = 0.068 \pm 0.013$ (1 s.d.), or ~18% relative uncertainty.*

*It is equally important to note that in our study, calculation of particle mass-concentration (eq. 4) does not explicitly incorporate estimates of $\varepsilon_{neb}$, but rather estimates of the extraction efficiency, ε (eq. 5), determined experimentally and independent of $\varepsilon_{neb}$. However, via eq. 6 (now included in the main text; see C5 below), ε is shown to be a function of $\varepsilon_{neb}$ and transmission efficiency, $\varepsilon_{trans}$. Since past studies (e.g., Cziczo et al., 2006) have illustrated that PALMS transmission is relatively stable, we thus*

take the uncertainty interval calculated for ε (~30% relative uncertainty at $\varepsilon(D_p = 746$ nm); eq. 5) to implicitly encapsulate uncertainties in nebulization efficiency.

*The above information has been included as supplementary material, and the following sentence added to Sect. 3.2.4 (Pg. 16 Lines 23-26):*

*"Note that while no systematic trends in nebulization drift were found over either hour-long measurement period, short term fluctuations in nebulization could occur; for the present experiment, such fluctuations are assumed to be encapsulated as uncertainty about the extraction efficiency parameterization (eq. 5; see Supplementary Material for details)."*

*Overall, we agree with the reviewer that future mass-concentration applications using SPMS – especially those where multiple successive samples are measured for mass concentration – should implement regular standardized checks for performance drift between samples. Text has also been added to 3.2.4 making this latter suggestion explicit (Pg. 17-18 Lines 25-29, 1-7; refer to C2 above).*

[Figure]

*Fig. R1. Results of three separate nebulization drift tests.*

C5) During the PSL calibration experiments, the authors used an OPS to monitor particle concentration and size to measure the transmission efficiency in to the PALMS (which was found

to be between 1 and – 16%). Why has this (or another method to measure size and number) not been done during the ice core sample analysis? By such, the transmission efficiency into the PALMS would have been measured using the real sample particles. Non-spherical shapes of the insoluble particles can certainly influence the transmission efficiency. Furthermore, methods exist to measure particle concentration in liquids. A comparison of the derived particle concentrations from the SPMS with such a reference measurement would have helped in validating the results. At least a determination of the mass concentration by filtering the solution and weighing the filter would have been possible, although here larger particle may dominate the total mass concentration. Without any comparison to an independent measurement of the same sample, the quantification of the results seems not reliable and overstretches the capabilities of an SPMS.

*We thank the reviewer for his/her suggestions, each of which is valid. We note that PALMS transmission capabilities are well-characterized, as validated by this and past studies (details found in Cziczo et al. (2006; 2013), thus the largest uncertainty likely arises from nebulization effects, as addressed above. More generally, as to the reviewer's primary concerns concerning mass concentration measurements, we iterate that this study's primary goal was not to "oversell" the use of SPMS for conducting mass concentration measurements of particles in snow and ice, but rather was in attempting to illustrate the feasibility of – including highlighting current fallbacks inherent in and the potential for future improvements to – the method. We hope that future studies may improve upon this baseline study by implementing comparisons to alternate well-founded techniques for measuring particle concentration in liquid (e.g., Coulter Counter techniques, UHSAS). These sentiments and suggestions have now been noted at the end of 3.2.4 (Pg. 17-18 Lines 26-29, 1-8):*

*"While our results show potential exists for using SPMS to determine insoluble mass concentrations of particles in snow and ice, they also identify areas where more work is needed before SPMS can be used as a quantitative tool. These include:*
*(…)*
*iv) comparing SPMS-derived particle concentrations with results from alternate, well-founded high-precision instrumentation (e.g., an Ultra-High Sensitivity Aerosol Spectrometer (UHSAS; Droplet Measurement Technologies Inc., Boulder, CO), or Coulter Counter instrumentation)."*

**Minor comments:**

Page 11, lines 1-3: How was determined that all water was evaporated?

*In general, PALMS will not provide spectra for particles that are water-saturated (i.e., "quenched"). However, for low-to-moderate degrees of water-saturation, PALMS can distinguish between wet and dry particles via water-cluster peaks (m/z = 18) that are prominent in the particle spectra (Murphy and Thompson, 1995). Determination of adequate water-evaporation could thus be experimentally determined by increasing the dry-to-wet flow ratio until evidence of water-saturation was eliminated in the particle spectra. The following text was added to Sect. 2.3 (Pg.'s 6-7 Lines 26, 1-2) clarifying this point:*

*"As PALMS generally does not provide spectra for water-saturated particles (Cziczo et al., 2006), the atomized particles were then adjoined with the dry flow, dropping the relative humidity and evaporating residual condensed water, resulting in dry residual particles entering the PALMS inlet."*

Figures of mass spectra in appendix: What is the method to select which mass peaks are labeled. Apparently not always the largest peaks? Are peaks unknown/not identified, e.g. in Fig. A1, lowest panel, around m/z 90? Additionally, minor tick marks and/or grid lines help the reader to determine the m/z number of a peak in the mass spectrum.

*In general, all major peaks corresponding to known ionic fragments are labeled. Specific details for this identification process can be found in prior studies, particularly references noted in Sect. 2.1 of the main text (e.g., Murphy and Thompson, 1997a,b; Cziczo et al., 2013). Many of the large peaks not labeled correspond to organic particles, or, in this study's case, particles that have undergone severe post-aqueous processing. There presently remains ambiguity in organic fragment identification, as is*

*the case for Fig. A1 for m/z ~ 90. The following text has been added (Pg. 20, Lines 15-17):*

*"All major peaks corresponding to known ionic fragments have been labeled (see Murphy and Thompson, 1997 a, b; and Cziczo et al., 2013)."*

*Grid lines and x-axis minor ticks have now been added to the (now Appendix 2's) mass-spectra plots.*

**References:**

*Cziczo, D. J., Froyd, K. D., Hoose, C., Jensen, E. J., Diao, M., Zondlo, M. A., Smith, J. B., Twohy, C. H. and Murphy, D. M.: Clarifying the Dominant Sources and Mechanisms of Cirrus Cloud Formation, Science, 340, 1320–1324, 2013.*

*Cziczo, D. J., Thomson, D. S., Thompson, T. L., DeMott, P. J., Murphy, D. M: Particle analysis by laser mass spectrometry (PALMS) studies of ice nuclei and other low number density particles, Int. J. Mass Spectrometry, 258, 21-29. 2006.*

*Murphy, D. M., and Thomson, D. S.: Chemical composition of single aerosol particles at Idaho Hill: Positive ion measurements, J. Geophys. Res., 102, 6353-6368, doi:10.1029/96JD00858, 1997a.*

*Murphy, D. M., and Thomson, D. S.: Chemical composition of single aerosol particles at Idaho Hill: Negative ion measurements, J. Geophys, Res., 102, 6353-6368. doi:10.1029/96JD00859, 1997b.*

*Murphy, D. M., and Thomson D. S.: Laser Ionization Mass Spectroscopy of Single Aerosol Particles, Aerosol Science and Technology, 22(3), 237-249, doi:10.1080/02786829408959743, 1995.*